# Stochastic Minimum-Cost Reach-Avoid Reinforcement Learning

**Jingduo Pan** [1 2]  **Taoran Wu** [1 2]  **Yiling Xue** [1 2 3]  **Bai Xue** [1 2 3]

## Abstract

We study stochastic minimum-cost reach-avoid reinforcement learning, where an agent must satisfy a reach-avoid specification with probability at least $p$ while minimizing expected cumulative costs in stochastic environments. Existing safe and constrained reinforcement learning methods typically fail to jointly enforce probabilistic reach-avoid constraints and optimize cost in the learning setting in stochastic environments. To address this challenge, we introduce reach-avoid probability certificates (RAPCs), which identify states from which stochastic reach-avoid constraints are satisfiable. Building on RAPCs, we develop a contraction-based Bellman formulation that serves as a principled surrogate for integrating reach-avoid considerations into reinforcement learning, enabling cost optimization under probabilistic constraints. We establish almost sure convergence of the proposed algorithms to locally optimal policies with respect to the resulting objective. Experiments in the MuJoCo simulator demonstrate improved cost performance and consistently higher reach-avoid satisfaction rates.

## 1. Introduction

Autonomous decision-making systems in safety-critical domains, such as robotic navigation and autonomous driving (Andrychowicz et al., 2020; Vagale et al., 2021), must satisfy safety constraints while efficiently accomplishing tasks. Many such problems naturally take the form of reach-avoid specifications, where an agent must reach a goal set while avoiding unsafe regions. Practical considerations such as energy consumption and control effort further motivate

minimizing cumulative trajectory cost (Vagale et al., 2021; Zhang et al., 2021; De Vries et al., 2024), leading to the minimum-cost reach-avoid problem (So et al., 2024).

In realistic environments, dynamics and observations are stochastic, making reach-avoid specifications inherently probabilistic. The stochastic minimum-cost reach-avoid problem therefore requires minimizing cumulative cost while ensuring the probability of reaching the goal without entering unsafe states exceeds a prescribed threshold, which remains challenging in reinforcement learning (Brunke et al., 2022). To motivate this, consider an Automated Guided Vehicle (AGV) navigating a warehouse (Fragapane et al., 2021). The AGV must reach a target loading dock while avoiding unpredictable collisions with human workers in a stochastic environment (Chen et al., 2019). The objective is to ensure that the probability of satisfying the reach-avoid constraint meets a safety threshold, while simultaneously minimizing cumulative battery consumption (De Ryck et al., 2020). This example captures the essence of our problem, where safety is specified as a probabilistic reach-avoid constraint and performance is measured by cumulative cost.

A large body of work in safe reinforcement learning studies the trade-off between task performance and safety constraints (Brunke et al., 2022), including trust-region, primal-dual, barrier-function, and reachability-inspired methods (Tessler et al., 2018; Chow et al., 2018a; Cheng et al., 2019; Yang et al., 2020; Zhang et al., 2020; Marvi & Kiumarsi, 2021; Yu et al., 2022; Ganai et al., 2023). However, in most approaches the reach objective is not imposed as an probabilistic specification, but is instead encoded implicitly through reward design, e.g., via sparse terminal rewards or dense shaping (Andrychowicz et al., 2017; Plappert et al., 2018; Trott et al., 2019; Liu et al., 2022). This scalarizes reach-avoid satisfaction into the return, blurring value semantics and making it difficult to jointly optimize cost efficiency and probabilistic reach-avoid satisfaction. A common workaround is to adopt surrogate formulations that retain this scalar trade-off structure, such as reward-cost scalarization or CMDP-style cumulative-cost constraints (Altman, 2021). However, such surrogates generally fail to preserve the structure of the original problem and require careful tuning of weights or thresholds, which could lead to infeasibility. More recently, RC-PPO (So et al., 2024) leverages Hamilton-Jacobi reachability to address minimum-cost

---

[1]Key Laboratory of System Software (Chinese Academy of Sciences), Institute of Software, Chinese Academy of Sciences, Beijing, China [2]University of Chinese Academy of Sciences, Beijing, China [3]School of Advanced Interdisciplinary Sciences, University of Chinese Academy of Sciences, Beijing, China. Correspondence to: Bai Xue <xuebai@ios.ac.cn>.

*Proceedings of the 43rd International Conference on Machine Learning*, Seoul, South Korea. PMLR 306, 2026. Copyright 2026 by the author(s).

reach-avoid but is restricted to deterministic settings.

For related work that explicitly enforces probabilistic constraint satisfaction, chance-constrained and risk-sensitive MDP formulations ensure safety by constraining the probability or tail risk of undesirable outcomes (Bäuerle & Jaśkiewicz, 2024). This line of work includes formulations that maximize target-reaching probability (Lin et al., 2003) and reinforcement learning methods based on CVaR (Chow et al., 2015) or quantile criteria (Li et al., 2022; Hau et al., 2024). However, these approaches typically characterize risk over accumulated returns rather than the probability of satisfying a reach-avoid specification over time, and thus lack a principled mechanism for jointly enforcing reach-avoid constraints while minimizing cumulative cost.

To address these limitations, we introduce the concept of the Reach-Avoid Probability Certificate (RAPC). An RAPC certifies that a given policy satisfies a reach-avoid specification with probability (i.e., reach-avoid probability) no less than a prescribed threshold. We develop a contraction-based Bellman formulation whose value function admits a clear probabilistic interpretation as a lower bound on the reach-avoid probability. When an RAPC exists, it provides a certificate of probabilistic task satisfaction. Building on this formulation, we propose Reach-Avoid Probability-Constrained Policy Optimization (RAPCPO). The method is motivated by this theoretically grounded value construction and incorporates a practical surrogate objective that empirically improves reach-avoid performance. We further show that RAPCPO converges almost surely to locally optimal policies of the surrogate objective under the proposed constrained optimization framework. We evaluate RAPCPO on a suite of stochastic minimum-cost reach-avoid benchmark tasks in the MuJoCo simulator. The experimental results demonstrate that, compared to state-of-the-art baselines, our approach achieves substantially lower cumulative cost while consistently satisfying the desired probabilistic reach-avoid specifications.

The main contributions of this paper are summarized below:

- We present a contraction-based Bellman formulation for stochastic reach-avoid problems that encodes both safety and liveness requirements, and introduce Reach-Avoid Probability Certificates as a principled mechanism for enforcing probabilistic reach-avoid constraints.

- We propose RAPCPO, a reinforcement learning algorithm for stochastic minimum-cost reach-avoid problems, and establish almost-sure convergence to locally optimal policies under the proposed probabilistic constraint framework.

- We demonstrate the effectiveness of the proposed approach on challenging high-dimensional control tasks,

achieving substantial improvements in cost efficiency while consistently achieving high satisfaction rates for the desired probabilistic reach-avoid specifications.

## 2. Related Works

**Safe and risk-sensitive reinforcement learning.** Safe reinforcement learning methods commonly balance task performance and safety using reward shaping, penalties, or constrained optimization (Altman, 2021; Brunke et al., 2022). In many approaches, reach objectives are encoded implicitly through sparse terminal rewards or dense shaping (Andrychowicz et al., 2017; Trott et al., 2019), which entangles goal satisfaction with cost accumulation and obscures the semantics of the learned value functions. Constrained Markov decision processes (CMDPs) instead impose constraints on cumulative costs, with deep RL solutions based on trust-region or primal-dual methods (Achiam et al., 2017; Yang et al., 2020). Related risk-sensitive and chance-constrained formulations explicitly reason about uncertainty by constraining probabilities or tail risks of undesirable outcomes (Bäuerle & Jaśkiewicz, 2024), including maximizing target-reaching probability (Lin et al., 2003) and methods based on CVaR or quantile criteria (Chow et al., 2015; Li et al., 2022). However, However, these methods typically characterize risk over accumulated returns or rely on surrogate constraints, which do not explicitly capture the joint structure of stochastic minimum-cost reach-avoid problems and do not directly encode probabilistic reach-avoid satisfaction. In contrast, we develop a Bellman-based value formulation that admits a clear probabilistic interpretation as a lower bound on reach-avoid probability. This formulation underpins Reach-Avoid Probability Certificates (RAPCs) and guides the design of a surrogate objective for policy optimization.

**Formal Specifications and Reachability-Based Methods.** Barrier- and Lyapunov-based methods provide formal safety and stability guarantees in control and safe RL (Khalil, 2002; Berkenkamp et al., 2017; Chow et al., 2018b). For reach-avoid tasks, representative approaches include CLBF-based methods (Ames et al., 2019) and barrier-like methods (Xue, 2026). Similarly, HJ reachability offers principled, formally correct reach-avoid solutions (Tomlin et al., 2000), and has been connected to RL (Fisac et al., 2019; Yu et al., 2022; Ganai et al., 2023). More broadly, reach-avoid requirements can be viewed as a special case of Linear Temporal Logic (LTL) specifications, motivating a rich line of RL literature on learning under complex formal specifications (Le et al., 2024; Svoboda et al., 2024). Despite their theoretical rigor, however, these families of methods share a fundamental limitation: they focus primarily on specification satisfaction or maximizing satisfaction probability, rather than optimizing performance objectives such as cumulative cost. Recently,

RC-PPO (So et al., 2024) addresses minimum-cost reach-avoid using HJ reachability, but is limited to deterministic environments. In contrast, we study stochastic environments and develop a Bellman-based surrogate framework for policy optimization under probabilistic reach-avoid objectives.

# 3. Preliminary

We briefly introduce Markov Decision Processes (MDPs) and formulate the stochastic minimum-cost reach-avoid problem considered in this paper.

## 3.1. Markov Decision Processes

We consider a discounted MDP $\mathcal{M} = (\mathcal{X}, \mathcal{A}, P, c, g, h, \gamma)$ with state space $\mathcal{X} \subset \mathbb{R}^n$ and action space $\mathcal{A} \subset \mathbb{R}^m$. The transition kernel $P(\cdot \mid x, a)$ specifies the distribution of the next state given $(x, a)$, $c : \mathcal{X} \times \mathcal{A} \to \mathbb{R}$ is a bounded stage cost, and $\gamma \in (0, 1)$ is the discount factor. We define $1_B(x) = 1_{x \in B} = 1$ if $x \in B$, and 0 otherwise. In addition, we assume that $\mathcal{X}$ is forward invariant, that is, $P(\mathcal{X} \mid x, a) = 1, \forall (x, a) \in \mathcal{X} \times \mathcal{A}$. This assumption is widely used in the reinforcement learning and stochastic control literature (Ganai et al., 2023; Žikelić et al., 2023). It ensures that trajectories starting from $\mathcal{X}$ remain in $\mathcal{X}$ almost surely, so that the value-function-type expectations appearing in (8) and (9) are well defined throughout the analysis. Without such an invariance property, the next state may leave $\mathcal{X}$ with nonzero probability, and additional conventions would be needed to define quantities outside the state space. We refer the reader to (Cao et al., 2025) for more details.

We define a target set $\mathcal{T} \subset \mathcal{X}$ and an unsafe set $\mathcal{F} \subset \mathcal{X}$, assumed to be disjoint. To encode reach and safety, define bounded functions $g, h : \mathcal{X} \to \mathbb{R}$ with $M > 0$: Let $M > 0$. Define the functions

$$g(x) := \begin{cases} -M, & x \in \mathcal{T}, \\ > 0, & x \notin \mathcal{T}, \end{cases} \quad h(x) := \begin{cases} M, & x \in \mathcal{F}, \\ -M, & x \notin \mathcal{F}. \end{cases} \quad (1)$$

A stochastic policy $\pi(\cdot|x)$ induces trajectories $(x_t, a_t)_{t \geq 0}$ with $a_t \sim \pi(\cdot|x_t)$ and $x_{t+1} \sim P(\cdot|x_t, a_t)$. The state-action value function $Q_c^\pi(x, a) = \mathbb{E}_{\pi, P}[\sum_{t=0}^\infty \gamma^t c(x_t, a_t) \mid x_0 = x, a_0 = a]$. The state-value function $V_c^\pi(x) = \mathbb{E}_{a \sim \pi}[Q_c^\pi(x, a)]$ satisfies the recursive Bellman relation: $V_c^\pi(x) = \mathbb{E}_{a \sim \pi}[c(x, a) + \gamma \mathbb{E}_{x' \sim P}[V_c^\pi(x')]]$. In addtion, given an initial state distribution $\rho$, $d_\rho^\pi$ denotes the steady-state distribution induced by policy $\pi$ (Bojun, 2020).

## 3.2. Problem Formulation

In stochastic systems, we quantify reach-avoid task satisfaction using probabilities.

**Definition 3.1** (Reach-Avoid Event and Probability)**.** For

a trajectory $\tau = (x_t, a_t)_{t \geq 0}$ with $x_0 = x$, define the hitting time $T(\tau) = \inf\{t \geq 0 : x_t \in \mathcal{T}\}$. We define the reach-avoid event $\mathbf{RA}_x$ as the set of successful trajectories:

$$\mathbf{RA}_x = \{\tau \mid T(\tau) < \infty, \ \{x_t\}_{t=0}^{T(\tau)} \cap \mathcal{F} = \emptyset\}. \quad (2)$$

The reach-avoid probability is defined as $\mathbb{P}_\pi(\mathbf{RA}_x) = \mathbb{E}_\pi[1_{\tau \in \mathbf{RA}_x} \mid x_0 = x]$. For any random variable $Z$, when $\mathbb{P}_\pi(\mathbf{RA}_x) > 0$, its conditional expectation is defined as:

$$\mathbb{E}_\pi[Z \mid \mathbf{RA}_x] = \frac{\mathbb{E}_\pi[Z \cdot 1_{\tau \in \mathbf{RA}_x} \mid x_0 = x]}{\mathbb{P}_\pi(\mathbf{RA}_x)}. \quad (3)$$

Base on Definition 3.1, we define the p-Reach-Avoid set $\mathbf{RA}_p$ as follows.

**Definition 3.2** ($p$-Reach-Avoid set)**.**

$$\mathcal{X}_p^\pi := \{x \in \mathcal{X} \mid \mathbb{P}_\pi(\mathbf{RA}_x) \geq p\}, \quad (4)$$

where $p \in (0, 1)$ is a user-specified probability threshold and $\pi$ denotes the policy under consideration. A policy is feasible if $\mathcal{X}_p^\pi \neq \emptyset$. The largest $p$-Reach-Avoid set $\mathcal{X}_p \subset \mathcal{X}$ is composed of states $x$ from which there exists at least one policy that keeps the system satisfying $\mathbb{P}_\pi(\mathbf{RA}_x) \geq p$, i.e.,

$$\mathcal{X}_p := \{x \in \mathcal{X} \mid \exists \pi \text{ s.t. } \mathbb{P}_\pi(\mathbf{RA}_x) \geq p\}. \quad (5)$$

Practical reach-avoid tasks require the reach-avoid probability to exceed a prescribed threshold $p$. Accordingly, our objective is twofold: over the feasible set $\mathcal{X}_p$, the policy minimizes cost while satisfying the reach-avoid constraint, whereas outside $\mathcal{X}_p$ it prioritizes improving reach-avoid probability. We formulate the resulting minimum-cost reach-avoid problem as follows:

$$\min_\pi \ \mathbb{E}_{x, a \sim d_\rho^\pi} \left[ V_c^\pi(x) \cdot 1_{\mathcal{X}_p}(x) - \mathbb{P}_\pi(\mathbf{RA}_x) \cdot 1_{\mathcal{X} \setminus \mathcal{X}_p}(x) \right]$$
$$\text{s.t. } \forall x \in \mathcal{X}_p, \ \mathbb{P}_\pi(\mathbf{RA}_x) \geq p. \quad (6)$$

# 4. Reach-Avoid Probability Certificates

This section introduces the reach-avoid probability certificate (RAPC), defined as a function that lower-bounds the exact reach-avoid probability under a fixed policy. The definition of an RAPC is intentionally broad: it requires only that the certificate constitute a valid lower bound, while subsequent results provide constructive and verifiable sufficient conditions based on Bellman-type equations.

**Reach-avoid probability certificates.** Since computing the exact reach-avoid probability $\mathbb{P}_\pi(\mathbf{RA}_{x_0})$ is generally intractable, a common approach is to derive computable lower bounds by searching for a function that certifies feasibility (Xue et al., 2021; Žikelić et al., 2023; Xue, 2026).

**Definition 4.1** (Reach-Avoid Probability Certificate (RAPC))**.** Given a policy $\pi$ and a state $x_0 \in \mathcal{X}$, a bounded function $v^\pi : \mathcal{X} \to \mathbb{R}$ is called a RAPC if it satisfies

$$\mathbb{P}_\pi(\mathbf{RA}_{x_0}) \geq v^\pi(x_0). \tag{7}$$

RAPC provides a rigorous lower bound of $\mathbb{P}_\pi(\mathbf{RA}_{x_0})$. In the sequel, we present two conditions that yield RAPCs: (i) an indicator-based formulation from (Xue, 2026), and (ii) a new enhanced Bellman equation that is more amenable to learning in black-box environments.

## 4.1. Indicator-Based Characterization

In (Xue, 2026), a necessary and sufficient condition was established to certify whether $\mathbb{P}_\pi(\mathbf{RA}_{x_0})$ exceeds a pre-scribed threshold $p$ under Assumption 4.2.

**Assumption 4.2.** For $x_0 \in \mathcal{X}$ and $\pi$, the reach-avoid probability satisfies $\mathbb{P}_\pi(\mathbf{RA}_{x_0}) > p$.

**Proposition 4.3.** *If there exist a bounded function $v^\pi_\gamma : \mathcal{X} \to \mathbb{R}$ and $\gamma \in (0,1)$ satisfying*

$$\begin{cases} v^\pi_\gamma(x_0) \geq p, \\ v^\pi_\gamma(x) = 1_\mathcal{T}(x) + \\ \quad \gamma 1_{\mathcal{X} \setminus (\mathcal{T} \cup \mathcal{F})}(x) \mathbb{E}_{a \sim \pi, x' \sim P(x'|x,a)} \left[ v^\pi_\gamma(x') \right], \end{cases} \tag{8}$$

*then $\mathbb{P}_\pi(\mathbf{RA}_{x_0}) \geq p$. Moreover, $v^\pi_\gamma$ is a RAPC. On the other hand, under Assumption 4.2, there exist a bounded function $v^\pi_\gamma : \mathcal{X} \to \mathbb{R}$ and $\gamma \in (0,1)$ satisfying (8).*

Despite its theoretical appeal, (8) poses practical challenges as follows. First, jointly learning $v^\pi_\gamma$ and $\gamma$ without explicitly enforcing $v^\pi_\gamma(x_0) \geq p$ admits degenerate solutions (e.g., $\gamma \to 0$) that satisfy the Bellman equation but drive $v^\pi_\gamma$ to zero for $x \in \mathcal{X} \setminus \mathcal{T}$, yielding an uninformative reach-avoid estimate. Second, even with a fixed discount factor $\gamma$, approximating the value function $v^\pi_\gamma$ using TD-based algorithms suffers from inherently sparse learning signals: outside the target set $\mathcal{T}$, the immediate term vanishes, and informative feedback is obtained only upon reaching the target. Moreover, when (8) provides insufficient learning signals, an agent optimizing the cost objective may converge to degenerate behaviors (e.g., remaining near the initial region) rather than actively exploring trajectories that satisfy the reach–avoid specification.

## 4.2. An Enhanced Bellman Formulation

To overcome the sparse learning signal of (8) while retaining a valid probabilistic certificate, we leverage the shaped functions $g(\cdot)$ and $h(\cdot)$ from Section 3. These functions define a max-min clamped Bellman operator that encodes reach-avoid constraints via boundary values, ensuring its unique fixed point admits a reach-avoid probability certificate. Choosing $g(x)$ to vary over $\mathcal{X} \setminus \mathcal{T}$ provides denser

signals away from the target, improving sample efficiency in black-box environments.

We introduce the following Bellman operator and prove the following lemma in Appendix A.1.

**Lemma 4.4.** *Let $\gamma \in (0,1)$ and let $V : \mathcal{X} \to \mathbb{R}$ be any bounded function. Define the operator $B^\pi[\cdot]$ by*

$$B^\pi[V](x) := \max \left\{ \begin{matrix} h(x), \min \left\{ g(x), \right. \\ \gamma \mathbb{E}_{a \sim \pi, \, x' \sim P(\cdot|x,a)} \left[ V(x') \right] \right\} \end{matrix} \right\}. \tag{9}$$

*Then $V(x) = B^\pi[V](x)$ admits a unique bounded solution $V^\pi_{g,h}$, and $B^\pi$ is a $\gamma$-contraction under $\| \cdot \|_\infty$. Moreover, we have the state-action value $Q^\pi_{g,h}(s,a) = \max \left\{ h(x), \min \left\{ g(x), \gamma \cdot \mathbb{E}_{x' \sim P(\cdot|x,a)} \left[ V^\pi_{g,h}(x') \right] \right\} \right\}$.*

Then the following result holds and the proof is given in Appendix A.2.

**Theorem 4.5.** *Let $V^\pi_{g,h}$ be the unique bounded fixed point of (9). Then for any initial state $x \in \mathcal{X}$, if $V^\pi_{g,h}(x) < 0$, then*

$$\mathbb{P}_\pi(\mathbf{RA}_x) \; \geq \; -\frac{1}{M} V^\pi_{g,h}(x). \tag{10}$$

*Consequently, if $-\frac{1}{M} V^\pi_{g,h}(x_0) \geq p$, then $\mathbb{P}_\pi(\mathbf{RA}_{x_0}) \geq p$.*

**Interpretation and Compensation Factor.** We provide an intuitive interpretation of the fixed point $V^\pi_{g,h}$ by examining how different trajectory realizations contribute to the Bellman operator (9). Recall that $\mathbf{RA}_x$ denotes the set of trajectories that reach the target set $\mathcal{T}$ before entering the unsafe set, starting from $x_0 = x$.

While the Bellman fixed point $V^\pi_{g,h}$ is determined by all trajectories induced by policy $\pi$, only trajectories in $\mathbf{RA}_x$ exhibit a simple discounted recursion that directly links the value magnitude to the target hitting time. We therefore focus on this subset to expose the effect of discounting.

For any realization $\tau \in \mathbf{RA}_x$ with hitting time $T$, the operator (9) reduces along the trajectory to the linear recursion $V^\pi_{g,h}(x_t) = \gamma V^\pi_{g,h}(x_{t+1})$ for all $t < T$ because of the definitions of $g(\cdot)$ and $h(\cdot)$ as in Section 3, with boundary condition $V^\pi_{g,h}(x_T) = -M$, yielding $V^\pi_{g,h}(x_0) = -\gamma^T M$.

Isolating the contribution of such trajectories in $\mathbf{RA}_x$ leads to the following explanatory decomposition:

$$V^\pi_{g,h}(x) \approx \mathbb{E}_\pi \left[ -M\gamma^T \mid \mathbf{RA}_x, \, x_0 = x \right] \mathbb{P}_\pi(\mathbf{RA}_x). \tag{11}$$

This decomposition highlights a discount-induced attenuation effect: when the expected hitting time $T$ is large, the exponential decay of $\gamma^T$ suppresses the value magnitude even if the reach-avoid probability is high. This effect is

inherent to discounted formulations of long-horizon reach-avoid problems and cannot be eliminated by tuning $\gamma$ alone.

Since this attenuation arises exclusively from successful reach-avoid executions, we introduce a compensation factor that conditions on the event $\mathbf{RA}_x$.

**Definition 4.6** (Compensation Factor). For a policy $\pi$ and initial state $x_0 = x$, let $T$ denote the first hitting time of $\mathcal{T}$. The compensation factor is defined as

$$\phi_\gamma^\pi(x) := \mathbb{E}_\pi[\gamma^T \mid \mathbf{RA}_x, \, x_0 = x]. \tag{12}$$

Based on (11) and Theorem 4.5, $-V_{g,h}^\pi(x)/M$ induces a certified lower bound on the reach-avoid probability; however, in long-horizon tasks this bound can be significantly attenuated by discounting, since the Bellman recursion propagates value information backward with a factor of $\gamma^T$ along trajectories $\tau \in \mathbf{RA}_x$. Consequently, in long-horizon tasks, the magnitude $|V_{g,h}^\pi(x)|$ can be small even when the true reach-avoid probability is high, making the quantity $-V_{g,h}^\pi(x)/M$ overly conservative. This motivates the normalized surrogate. From the approximate decomposition in (11), we have $-\frac{V_{g,h}^\pi(x)}{M} \approx \phi_\gamma^\pi(x)\,\mathbb{P}_\pi(\mathbf{RA}_x)$, suggesting the normalized estimator

$$\hat{p}_\pi(x) \triangleq -\frac{V_{g,h}^\pi(x)}{M\,\phi_\gamma^\pi(x)}. \tag{13}$$

While $\hat{p}_\pi(x)$ is not a certified probability bound, it mitigates discount-induced attenuation and serves as an optimization surrogate. Removing $\phi_\gamma^\pi(x)$ leads to an overly conservative estimate and degrades performance, preventing the learned policy from achieving low costs (see in Section 6).

Therefore, given a probability threshold $p$ and a discount factor $\gamma$, we consider a surrogate optimization objective based on $-\frac{V_{g,h}^\pi(x)}{M\phi_\gamma^\pi(x)}$; since $M$ is a constant, it is omitted in the objective without affecting the optimizer. This leads to the following surrogate optimization problem (RAPCRL):

$$\min_\pi \quad \mathbb{E}_{x,a \sim d_\rho^\pi} \left[ V_c^\pi(x) \cdot 1_{\mathcal{X}_p} + \frac{V_{g,h}^\pi(x)}{\phi_\gamma^\pi(x)} \cdot 1_{\mathcal{X} \setminus \mathcal{X}_p} \right]$$
$$\text{s.t.} \quad \forall x \in \mathcal{X}_p, \quad V_{g,h}^\pi(x) \leq -M \cdot \phi_\gamma^\pi(x) \cdot p. \tag{14}$$

# 5. Stochastic Minimum-Cost Reach-Avoid Reinforcement Learning

This section proposes RAPCPO, an actor-critic framework for the surrogate stochastic minimum-cost reach-avoid problem (14). Unlike the certificate-based updates in Section 4, RAPCPO does not enforce reach-avoid feasibility at every iteration. Instead, it exploits certificate structure to construct policy-dependent surrogate learning signals that enable efficient optimization in long-horizon black-box environments. We also provide a theoretical analysis of the algorithm's convergence behavior.

---

**Algorithm 1** RAPCPO Actor-Critic

1: **Input:** MDP $M$; critic step size $\zeta_1(l)$; actor step size $\zeta_2(l)$; compensation-factor step size $\zeta_3(l)$; horizon $H$
2: **Initialize:** policy $\theta$; RAPC critic $\eta$; cost critic $\kappa$; compensation factor $\xi$.
3: **for** $l \leftarrow 0, 1, 2, \ldots$ **do**
4:     Initialize state $x_0 \sim \rho$
5:     **for** $t \leftarrow 0, 1, 2, \ldots, H - 1$ **do**
6:         Sample $a_t \sim \pi_\theta(\cdot \mid x_t)$; observe $x_{t+1}$, cost $c_t$, and signals $g(x_t), h(x_t)$; Store transition $(x_t, a_t, c_t, g(x_t), h(x_t), x_{t+1})$ in $\mathcal{D}$
7:         **RAPC critic update:** $\eta \leftarrow \eta - \zeta_1(l)\,\nabla_\eta \mathcal{J}_{Q_{g,h}}(\eta)$
8:         **Cost critic update:** $\kappa \leftarrow \kappa - \zeta_1(l)\,\nabla_\kappa \mathcal{J}_{Q_c}(\kappa)$
9:         Clamp $\phi_\gamma(x_t; \xi) \leftarrow \max\{\phi_\gamma(x_t; \xi), 10^{-6}\}$
10:         Compute $g_\theta$ in (23) using $1_{\mathcal{X}_p^{\pi_{\theta_l}}}(x_t)$ and Eqs. (15)-(19) (stop-grad through $1_{\mathcal{X}_p^{\pi_{\theta_l}}}(x_t), \phi_\gamma$)
11:         **Policy update:** $\theta \leftarrow \Gamma_\Theta\big(\theta - \zeta_2(l)\,g_\theta\big)$
12:     **end for**
13:     **Compensation-factor update:**
14:     **if** trajectory reaches $\mathcal{T}$ before entering $\mathcal{F}$ with first hitting time $T$ **then**
15:         Set $y_t = \gamma^{T-t}$ for all visited $x_t$ with $t < T$
16:         $\xi \leftarrow \xi - \zeta_3(l)\,\nabla_\xi \mathcal{J}_\phi(\xi)$
17:     **else**
18:         **skip** update of $\xi$
19:     **end if**
20: **end for**

---

## 5.1. Surrogate Optimization via Certificate-Guided Partitioning

We consider the surrogate optimization problem (14), which depends on the unknown $p$-reach-avoid set $\mathcal{X}_p$. In black-box environments, $\mathcal{X}_p$ is not explicitly available. Moreover, the compensation factor $\phi_\gamma^\pi(x)$ does not satisfy a Bellman equation, preventing direct application of standard policy gradient methods.

Rather than solving (14) directly, RAPCPO tackles a policy-dependent surrogate problem that heuristically approximates (14). This surrogate formulation is induced by reach-avoid critics, and the algorithm alternates between critic learning, surrogate-feasible set construction, and policy optimization with partitioned and normalized surrogate objectives. Let $\pi(x; \theta)$ denote the policy parameterized by $\theta$. At iteration $l$, given the current policy $\pi_{\theta_l}$ and critic estimates, we define the critic-induced surrogate-feasible set

$$\mathcal{X}_p^{\pi_{\theta_l}} = \left\{ x \in \mathcal{X} \,\middle|\, \begin{array}{l} V_{g,h}^{\pi_{\theta_l}}(x) \leq -pM\phi_\gamma^{\pi_{\theta_l}}(x), \\ \phi_\gamma^{\pi_{\theta_l}}(x) \geq 0 \end{array} \right\}. \tag{15}$$

The set $\mathcal{X}_p^{\pi_{\theta_l}}$ induces a policy-dependent partition of the state space, prioritizing cost optimization inside the set and

reach-avoid improvement outside. Using $\mathcal{X}_p^{\pi_{\theta_l}}$ and the current compensation-factor estimate $\phi_\gamma^{\pi_{\theta_l}}$, we guide the policy update at iteration $l$ by the surrogate problem

$$
\begin{aligned}
\min_\pi \quad & \mathbb{E}_{x,a\sim d_\rho^\pi}\left[V_c^\pi(x)\cdot 1_{\mathcal{X}_p^{\pi_{\theta_l}}}(x) + \frac{V_{g,h}^\pi(x)}{\phi_\gamma^{\pi_{\theta_l}}(x)}\cdot 1_{\mathcal{X}\setminus\mathcal{X}_p^{\pi_{\theta_l}}}(x)\right] \\
\text{s.t.} \quad & \forall x\in\mathcal{X}_p^{\pi_{\theta_l}}, \quad V_{g,h}^\pi(x)\le -M\cdot\phi_\gamma^{\pi_{\theta_l}}(x)\cdot p.
\end{aligned}
$$
(16)

During iteration $l$, $\phi_\gamma^{\pi_{\theta_l}}$ is treated as a fixed, non-differentiable normalization factor.

Consider the classical actor-critic framework with state-action value functions. We learn action-value critics $Q_{g,h}^\pi(x,a)$, $Q_c^\pi(x,a)$ and $\phi_\gamma^\pi(x)$ with function approximators $Q_{g,h}(x,a;\eta)$, $Q_c(x,a;\kappa)$ and $\phi_\gamma(x;\xi)$. The cost critic is updated via standard temporal-difference learning (Sutton et al., 1998) by minimizing the objective $\mathcal{J}_{Q_c}(\kappa)$; see Appendix B.1 for details. The reach-avoid critic is trained using the self-consistency condition in (9) by minimizing

$$
\mathcal{J}_{Q_{g,h}}(\eta) = \mathbb{E}_{(x,a)\sim\mathcal{D}}\left[\tfrac{1}{2}\left(e_{g,h}(x,a;\eta)\right)^2\right],
$$
(17)

where $e_{g,h}(x,a;\eta) := Q_{g,h}(x,a;\eta) - \hat{Q}_{g,h}(x,a)$ and

$$
\hat{Q}_{g,h}(x,a) = \max\left\{h(x), \min\left\{g(x),\right.\right.
$$
$$
\left.\left.\gamma\,\mathbb{E}_{x'\sim P(\cdot|x,a),\,a'\sim\pi(\cdot|x')}\left[Q_{g,h}(x',a';\eta)\right]\right\}\right\},
$$
(18)

$\mathcal{D}$ is the distribution of previously sampled states and actions (i.e., $d_\rho^\pi$), or a replay buffer, and $a$ is the action taken at $s$. The compensation factor $\phi_\gamma(x;\xi)$ is estimated from successful reach-avoid rollouts of the current policy. For trajectories that reach the target set $\mathcal{T}$ before the failure set $\mathcal{F}$, each visited state $x_t$ with hitting time $T$ is assigned the target value $y_t = \gamma^{T-t}$, and $\xi$ is updated by minimizing

$$
\mathcal{J}_\phi(\xi) = \mathbb{E}_{x_t\sim\mathcal{D}}\left[\left(\phi_\gamma(x_t;\xi) - y_t\right)^2\right].
$$
(19)

During policy optimization, $\phi_\gamma(x;\xi)$ is treated as a fixed, non-differentiable normalization factor.

Within $\mathcal{X}_p^{\pi_{\theta_l}}$, cost reduction and reach-avoid improvement may induce conflicting gradients. To mitigate such conflicts, we apply symmetric projection-based gradient rectification (Xu et al., 2021; Gu et al., 2024), which typically results in reach-avoid probabilities exceeding the prescribed threshold $p$ in RAPCPO, as each update jointly reduces the cost and increases the lower bound of the reach-avoid probability. Specifically, we compute three policy-gradient components:

$$
\begin{aligned}
g_r^{in} &= \frac{\nabla_\theta\mathbb{E}\left[Q_{g,h}^{\pi_\theta}(x,a;\eta)\,1_{\mathcal{X}_p^{\pi_{\theta_l}}}(x)\right]}{\max\{\phi_\gamma(x;\xi),10^{-6}\}}, \\
g_r^{out} &= \frac{\nabla_\theta\mathbb{E}\left[Q_{g,h}^{\pi_\theta}(x,a;\eta)\,1_{\mathcal{X}\setminus\mathcal{X}_p^{\pi_{\theta_l}}}(x)\right]}{\max\{\phi_\gamma(x;\xi),10^{-6}\}}, \\
g_c^{in} &= \nabla_\theta\mathbb{E}\left[Q_c^{\pi_\theta}(x,a;\kappa)\,1_{\mathcal{X}_p^{\pi_{\theta_l}}}(x)\right].
\end{aligned}
$$
(20)

Here the partition mask $1_{\mathcal{X}_p^{\pi_{\theta_l}}}(x)$ is treated as fixed within iteration $l$ (i.e., no gradients are propagated through the critic-induced set). When the in-set gradients are negatively aligned, i.e., $\langle g_r^{in}, g_c^{in}\rangle < 0$, we apply symmetric projection:

$$
\begin{aligned}
\tilde{g}_r^{in} &= g_r^{in} - \frac{\langle g_r^{in}, g_c^{in}\rangle}{\langle g_c^{in}, g_c^{in}\rangle}g_c^{in}, \\
\tilde{g}_c^{in} &= g_c^{in} - \frac{\langle g_c^{in}, g_r^{in}\rangle}{\langle g_r^{in}, g_r^{in}\rangle}g_r^{in}.
\end{aligned}
$$
(21)

The mixed in-set update direction is then

$$
g_{\mathrm{mix}} = \begin{cases} \tilde{g}_r^{in} + \tilde{g}_c^{in}, & \langle g_r^{in}, g_c^{in}\rangle < 0, \\ g_r^{in} + g_c^{in}, & \text{otherwise.} \end{cases}
$$
(22)

The final policy update direction is given by

$$
g_\theta = g_r^{out} + g_{\mathrm{mix}}.
$$
(23)

Algorithm 1 provides the pseudo-code of an actor-critic version of RAPCPO. The algorithm alternates between environment interaction and stochastic gradient updates of $\nabla_\eta\mathcal{J}_{Q_{g,h}}(\eta)$, $\nabla_\kappa\mathcal{J}_{Q_c}(\kappa)$, $\nabla_\xi\mathcal{J}_\phi(\xi)$, and the policy gradient $g$, with detailed derivations provided in Appendix B.1. In the algorithm, the $\Gamma_\Theta(\theta)$ operator projects a vector $\theta\in\mathbb{R}^k$ to the closest point in a compact and convex set $\Theta\subseteq\mathbb{R}^k$, i.e., $\Gamma_\Theta(\theta) = \arg\min_{\hat{\theta}\in\Theta}\|\hat{\theta} - \theta\|^2$.

## 5.2. Convergence Analysis

We analyze the convergence of RAPCPO in an actor–critic setting. Under standard conditions on step sizes, exploration, and bounded policy parameters, the policy iterates converge almost surely to the set of generalized stationary points of the surrogate objective, in the sense of differential inclusions. The formal result and proof are provided in Appendix B.2.

# 6. Experiments

Details of our on-policy RAPCPO implementation based on proximal policy optimization (PPO) (Schulman et al., 2017) are given in Appendix C, and we set $p = 0.5$ in RAPCPO. We evaluate RAPCPO to answer the following questions:

- Can RAPCPO achieve lower cost while maintaining a high reach-avoid probability in both deterministic and stochastic tasks?

- How critical is the compensation factor $\phi_\gamma^\pi$ to the performance of RAPCPO, as demonstrated through ablation studies?
- How does the parameter $p$ influence the trade-off between cost and reach-avoid probability?

**Benchmarks and Baselines.** Benchmark and implementation details are provided in Appendix D. We evaluate two environment classes: (i) deterministic minimum-cost reach-avoid tasks (PointGoal, FixedWing, Pendulum (Ray et al., 2019; So & Fan, 2023)); (ii) stochastic tasks (Safety Hopper, Safety HalfCheetah (Todorov et al., 2012)) with 10% Gaussian action noise. Across environments, the green region denotes the target set, while the red region indicates unsafe states.

We compare RAPCPO with representative baselines from three categories. CMDP-based methods with a predefined cost budget include RESPO (Ganai et al., 2023), Sauté (Sootla et al., 2022), and CPPO (Ying et al., 2022) (which constrains the CVaR of cumulative costs). Weighted-sum baselines optimize a scalarized reward; we include $PPO_\beta$ with a fixed Lagrange multiplier $\beta$. Reach-avoid-specific baselines include RC-PPO (So et al., 2024), designed for deterministic minimum-cost reach-avoid tasks.

**CMDP surrogate for evaluation.** Because the minimum-cost reach-avoid objective cannot be posed as a CMDP, we reformulate into the following surrogate CMDP:

$$
\begin{aligned}
\max_\pi \quad & \mathbb{E}_{\tau \sim \pi} \sum_t \left[ \gamma^t r(x_t, u_t) \right] \\
\text{s.t.} \quad & \begin{cases} \mathbb{E}_{\tau \sim \pi} \sum_t \left[ \gamma^t \mathbf{1}_{x_t \in \mathcal{F}} \times C_{\text{fail}} \right] \leq 0, \\ \mathbb{E}_{\tau \sim \pi} \sum_t \left[ \gamma^t c(x_t, u_t) \right] \leq \mathcal{X}_{\text{threshold}}. \end{cases}
\end{aligned}
\tag{24}
$$

where the reward $r$ incentivies goal-reaching, $C_{\text{fail}}$ is a term balancing two constraint terms, and we set $\mathcal{X}_{\text{threshold}}$ to the average minimum cumulative cost achieved by RAPCPO over multiple runs, yielding comparable cost tolerance across methods. For $PPO_\beta$, we use the modified reward $r(x_t) - \beta \left( \mathbf{1}_{x_t \in \mathcal{F}} C_{\text{fail}} + c(x_t, u_t) \right)$. All baseline methods incorporate distance-based potential reward shaping for learning. As shown in (So et al., 2024), CMDP surrogates and weighted-sum objectives are structurally misaligned with the minimum-cost reach-avoid objective, leading to an inherent performance gap. RAPCPO avoids this mismatch by optimizing the reach-avoid objective directly, and our experiments validate the resulting improvement over strong baselines.

### 6.1. Deterministic Minimum-Cost Reach-Avoid

Figure 1 illustrates the deterministic benchmark environments. Across PointGoal and FixedWing, RAPCPO consistently achieves the lowest cumulative cost while maintaining

higher or comparable reach rates relative to $PPO_\beta$, Sauté, CPPO, and RESPO (Figure 2). Sauté and CPPO impose extremely stringent constraint handling when solving the optimization problem. Such treatment is effective for avoiding unsafe regions, where the constraint is only triggered by a small subset of states. However, when the cost is incurred at every state, Saute and CPPO with CVaR constraints become overly conservative, leading to severely degraded performance and even extremely low reach–avoid task success rates. In contrast, expectation-based constraints yield substantially better empirical results in this setting.

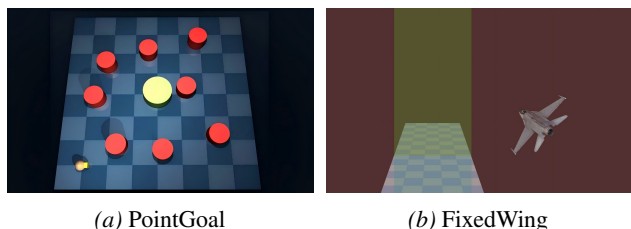

*(a)* PointGoal      *(b)* FixedWing

*Figure 1.* Deterministic minimum-cost reach-avoid benchmark environments.

Under the same number of environment-interaction iterations, RC-PPO often yields inferior performance, as summarized in Table 1. To ensure a fair comparison, we additionally report RC-PPO results trained with twice the number of iterations in Figure 2.

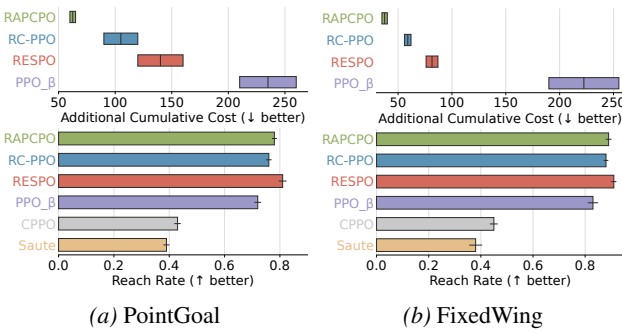

*(a)* PointGoal      *(b)* FixedWing

*Figure 2.* Cumulative cost and reach rate on deterministic benchmarks. RAPCPO achieves lower cost with competitive or higher reach rates compared to baselines.

*Table 1.* Reach rate comparison between RC-PPO and RAPCPO under the same iteration budget.

| Method | PointGoal | FixedWing |
|---|---|---|
| RC-PPO | 62.29% | 73.98% |
| RAPCPO (ours) | **78.49%** | **88.67%** |

On Pendulum, RC-PPO and RAPCPO exhibit similar energy-pumping behaviors and achieve comparable reach

rates and accumulated cost. We therefore focus on comparing RAPCPO against RESPO to highlight the impact of explicitly optimizing minimum cost. As shown in Figure 3, RAPCPO achieves lower cumulative cost by exploiting longer, cost-efficient trajectories.

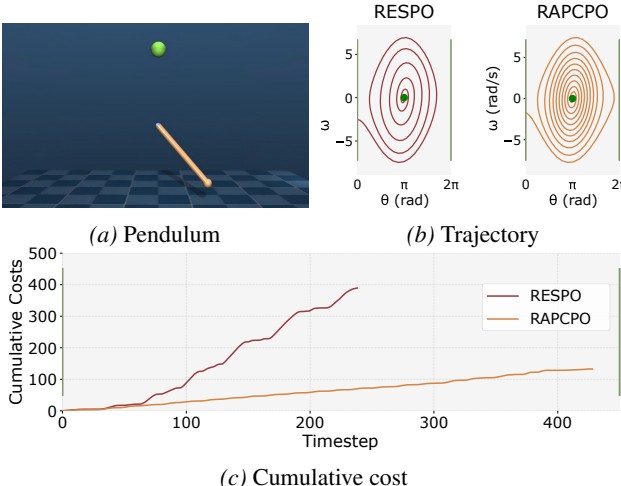

*(a)* Pendulum          *(b)* Trajectory

*(c)* Cumulative cost

*Figure 3.* Pendulum results. RAPCPO achieves lower cumulative cost by favoring longer, smoother trajectories.

## 6.2. Stochastic Minimum-Cost Reach-Avoid

Figure 4 illustrates the stochastic benchmark environments. Under action noise, RAPCPO consistently outperforms all baselines on both Safety Hopper and Safety HalfCheetah (Figure 5). In particular, RAPCPO achieves substantially lower cumulative cost while maintaining higher reach rates.

As in the deterministic setting, Sauté and CPPO with CVaR constraints are overly conservative when costs are statewise, leading to poor performance, while expectation-based constraints perform significantly better. And under the same iteration budget, RC-PPO similarly exhibits instability. We therefore additionally report RC-PPO trained with twice the number of iterations.

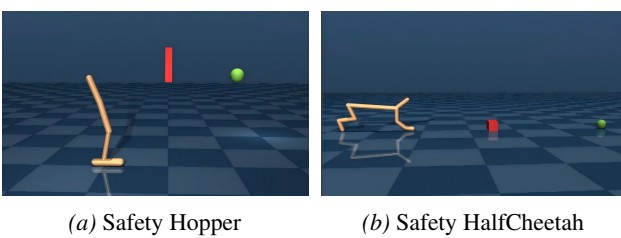

*(a)* Safety Hopper          *(b)* Safety HalfCheetah

*Figure 4.* Stochastic minimum-cost reach-avoid benchmark environments.

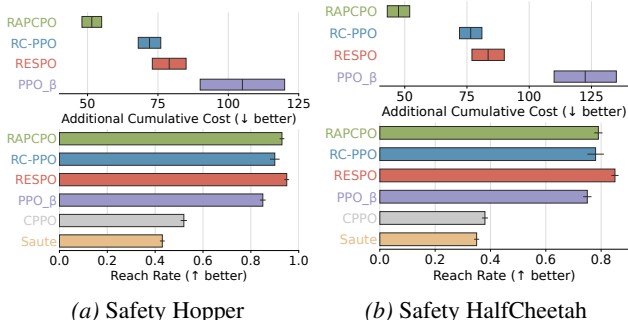

*(a)* Safety Hopper          *(b)* Safety HalfCheetah

*Figure 5.* Cumulative cost and reach rate on stochastic benchmarks. RAPCPO achieves the best cost-success trade-off among all methods.

## 6.3. Ablation Study: Compensation Factor $\phi_\gamma^\pi$ and Hyperparameter $p$

We isolate the auxiliary compensation factor on Frozen-Lake (Brockman et al., 2016), where the objective is to maximize the probability of safely reaching the goal. Figure 8 in Appendix D.2 compares the true reach-avoid probability with estimates obtained using the fixed-$\gamma$ Bellman equation in (8), with and without the compensation factor. Incorporating the compensation factor yields a substantially less conservative and better-calibrated approximation.

We further evaluate the compensation factor on minimum-cost reach-avoid tasks using the enhanced Bellman formulation (9). As shown in Figure 6, incorporating the compensation factor achieves lower cumulative cost while maintaining a sufficiently high reach rate across tasks.

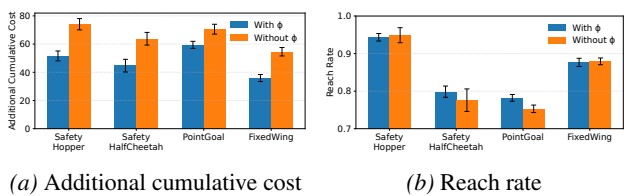

*(a)* Additional cumulative cost      *(b)* Reach rate

*Figure 6.* Impact of the compensation factor on cost and reach rate across tasks.

Then, when both methods employ the compensation factor, we compare the fixed-$\gamma$ Bellman formulation (8) with the enhanced Bellman formulation (9) for minimum costs reach-avoid problem. Under the same iteration budget, the fixed-$\gamma$ Bellman formulation (8) suffers from severe reward sparsity and achieves low reach rates. In contrast, the enhanced formulation substantially improves task completion across all environments (Table 2).

Finally, Figure 7 analyzes the effect of the hyperparameter p in the Safety Hopper environment. When $p = 0$, the reach-related signal is too weak to guide the algorithm toward policies satisfying the reach-avoid constraints, and

*Table 2.* Reach rate comparison between the fixed-$\gamma$ Bellman formulation and the enhanced Bellman formulation under the same iteration budget.

| Method | Safety HalfCheetah | Safety Hopper | PointGoal | FixedWing |
|---|---|---|---|---|
| Fixed-$\gamma$ Bellman | 0.4387 | 0.3218 | 0.4526 | 0.4713 |
| Enhanced Bellman | **0.7986** | **0.9434** | **0.7821** | **0.8764** |

the minimum-cost objective dominates, leading to near-stationary behaviors with very low reach rate and minimal cost. For $p$ in the range $[0.1, 0.7]$, sufficient reach-related signal is provided, enabling the learned policies to achieve high reach rates while keeping the additional cumulative cost moderate. However, due to the 10% action noise in the Safety Hopper environment, larger $p$ values 0.8-1.0 are overly aggressive and bias the optimization toward reach rate at the expense of cost minimization. As shown in the cost plot (log scale), this results in a sharp increase in additional cumulative cost with only marginal gains in reach rate.

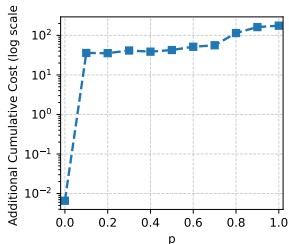 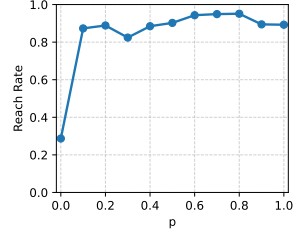

    *(a)* Cumulative cost(log scale)      *(b)* Reach rate

*Figure 7.* Ablation on the hyperparameter p in the Safety Hopper environment. Right: reach rate as a function of $p$. Left: additional cumulative cost as a function of $p$. The cost is plotted on a logarithmic scale due to its wide dynamic range.

## 7. Conclusion and Limitations

This paper addressed the minimum-cost reach-avoid reinforcement learning problem in stochastic environments. We first proposed the Reach-Avoid Probability Certificate, and then established sufficient conditions within a Bellman recursive framework to ensure the existence of such certificates meeting the required threshold. Leveraging these theoretical foundations, we develop RAPCPO, a reinforcement learning algorithm inspired by this formulation and guided by a practical surrogate objective that empirically improves reach-avoid performance. We further show that RAPCPO converges almost surely to locally optimal policies under the proposed framework. Experimental evaluations show that RAPCPO significantly reduces both task costs and failure rates, demonstrating superior performance compared to existing baseline methods. Despite its advantages, RAPCPO also has inherent limitations. Notably, Theorem 4.5 provides only a sufficient condition, rather than a necessary and

sufficient one. In future work, we plan to explore new conditions that are both sufficient and necessary, while remaining computationally tractable.

## Acknowledgements

The authors would like to thank anonymous reviewers and meta-reviewers for their insightful comments. This work was partially supported by the CAS Pioneer Hundred Talents Program and the Basic Research Program of the Institute of Software, Chinese Academy of Sciences (Grant No. ISCAS-JCMS-202302).

## Impact Statement

This paper presents work whose goal is to advance reinforcement learning under probabilistic reach-avoid constraints. While the proposed methods may have potential applications in decision-making under uncertainty, this work focuses on algorithmic development and simulation-based evaluation and does not raise ethical concerns beyond those commonly associated with reinforcement learning research.

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

# A. Proofs

## A.1. The proof for lemma 4.4

*Proof.* Let $\mathcal{B}(\mathbb{R}^n)$ denote the space of all bounded functions mapping from $\mathbb{R}^n$ to $\mathbb{R}$. That is,

$$\mathcal{B}(\mathbb{R}^n) = \left\{ V : \mathbb{R}^n \to \mathbb{R} \mid \sup_{x \in \mathbb{R}^n} |V(x)| < \infty \right\}.$$

It is a standard result that the space $\mathcal{B}(\mathbb{R}^n)$ of bounded functions, endowed with the supremum norm $\|V\|_\infty = \sup_{x \in \mathbb{R}^n} |V(x)|$, constitutes a Banach space (Folland, 1999).

First, we verify that $B^\pi$ maps $\mathcal{B}(\mathbb{R}^n)$ to itself. Let $V \in \mathcal{B}(\mathbb{R}^n)$ with $\|V\|_\infty \leq M$. Since $h$ and $g$ are assumed bounded, let $\|h\|_\infty \leq M_h$ and $\|g\|_\infty \leq M_g$. Let $E[V](x) = \gamma \cdot \mathbb{E}_{a \sim \pi, x' \sim P(\cdot|x,a)}[V(x')]$. We have:

$$|E[V](x)| = |\gamma \mathbb{E}[V(x')]| \leq \gamma \mathbb{E}[|V(x')|] \leq \gamma \mathbb{E}[\|V\|_\infty] = \gamma \|V\|_\infty \leq \gamma M.$$

The term $\min\{g(x), E[V](x)\}$ is therefore bounded:

$$|\min\{g(x), E[V](x)\}| \leq \max\{|g(x)|, |E[V](x)|\} \leq \max\{M_g, \gamma M\}.$$

Finally, $B^\pi[V](x) = \max\{h(x), \min\{g(x), E[V](x)\}\}$ is bounded:

$$|B^\pi[V](x)| \leq \max\{|h(x)|, |\min\{g(x), E[V](x)\}|\} \leq \max\{M_h, \max\{M_g, \gamma M\}\}.$$

Since the bound does not depend on $x$, $\|B^\pi[V]\|_\infty$ is finite, and thus $B^\pi[V] \in \mathcal{B}(\mathbb{R}^n)$.

Next, we prove the contraction property. Let $V_1, V_2 \in \mathcal{B}(\mathbb{R}^n)$. For any $x \in \mathbb{R}^n$:

$$\begin{aligned}
|B^\pi[V_1](x) - B^\pi[V_2](x)| &= |\max\{h(x), \min\{g(x), E[V_1](x)\}\} - \max\{h(x), \min\{g(x), E[V_2](x)\}\}| \\
&\leq |\min\{g(x), E[V_1](x)\} - \min\{g(x), E[V_2](x)\}| \\
&\quad \text{(Using the property } |\max(a,c) - \max(b,c)| \leq |a-b|) \\
&\leq |E[V_1](x) - E[V_2](x)| \\
&\quad \text{(Using the property } |\min(a,c) - \min(b,c)| \leq |a-b|) \\
&= \left| \gamma \mathbb{E}_{a \sim \pi, x' \sim P(\cdot|x,a)}[V_1(x')] - \gamma \mathbb{E}_{a \sim \pi, x' \sim P(\cdot|x,a)}[V_2(x')] \right| \\
&= \gamma \left| \mathbb{E}_{a \sim \pi, x' \sim P(\cdot|x,a)}[V_1(x') - V_2(x')] \right| \quad \text{(Linearity of Expectation)} \\
&\leq \gamma \mathbb{E}_{a \sim \pi, x' \sim P(\cdot|x,a)}[|V_1(x') - V_2(x')|] \quad \text{(Using } |\mathbb{E}[X]| \leq \mathbb{E}[|X|]) \\
&\leq \gamma \mathbb{E}_{a \sim \pi, x' \sim P(\cdot|x,a)}[\|V_1 - V_2\|_\infty] \quad \text{(Definition of sup norm)} \\
&= \gamma \|V_1 - V_2\|_\infty \cdot \mathbb{E}_{a \sim \pi, x' \sim P(\cdot|x,a)}[1] \\
&= \gamma \|V_1 - V_2\|_\infty. \quad \text{(Expectation of constant is constant)}
\end{aligned}$$

Since the inequality $|B^\pi[V_1](x) - B^\pi[V_2](x)| \leq \gamma \|V_1 - V_2\|_\infty$ holds for all $x \in \mathbb{R}^n$, we can take the supremum over $x$ on the left side:

$$\|B^\pi[V_1] - B^\pi[V_2]\|_\infty = \sup_{x \in \mathbb{R}^n} |B^\pi[V_1](x) - B^\pi[V_2](x)| \leq \gamma \|V_1 - V_2\|_\infty.$$

Given that $\gamma \in (0, 1)$, the operator $B^\pi$ is a contraction mapping on the Banach space $(\mathcal{B}(\mathbb{R}^n), \|\cdot\|_\infty)$.

By the Banach Fixed-Point Theorem, a contraction mapping on a complete metric space has a unique fixed point. Therefore, there exists a unique function $V_{g,h}^\pi \in \mathcal{B}(\mathbb{R}^n)$ such that $V_{g,h}^\pi = B^\pi[V_{g,h}^\pi]$. $\square$

## A.2. The proof for Theorem 4.5

Firstly, we introduce a proposition; the proof can be found in (Xue, 2026).

**Proposition A.1.** *Under Assumption 2, there exist a constant $\gamma \in (0, 1)$ and a function $v(x) : \mathbb{R}^n \to \mathbb{R}$, which is bounded over $\mathcal{X}$ and satisfies the following condition:*

$$\begin{cases}
v(x_0) \geq \epsilon_2, \\
v(x) \leq \gamma \, \mathbb{E}_\theta[v(f(x, \theta))], & \forall x \in \mathcal{X} \setminus \mathcal{X}_r, \\
v(x) \leq 1, & \forall x \in \mathcal{X}_r, \\
v(x) \leq 0, & \forall x \in \mathbb{R}^n \setminus \mathcal{X},
\end{cases} \tag{25}$$

*if and only if* $\mathbb{P}_\pi(RA_{x_0}) \geq \epsilon_2$.

Secondly, we prove that $V_{g,h}^\pi(x)$ satisfying the following conditions:

$$\begin{cases} V_{g,h}^\pi(x) \geq -M, & \text{if } x \in \mathcal{T}, \\ V_{g,h}^\pi(x) \geq 0, & \text{if } x \in \mathcal{F}, \\ V_{g,h}^\pi(x) \geq \gamma \mathbb{E}_{a \sim \pi(\cdot|x), x' \sim P(\cdot|x,a)}[V_{g,h}^\pi(x')], & \text{if } x \in \{\mathcal{X} \setminus (\mathcal{T} \cup \mathcal{F})\} \cap \{x | V_{g,h}^\pi(x) < 0\}. \end{cases} \tag{26}$$

*Proof.* By the definitions of $g$, $h$, and $V_{g,h}^\pi(x)$, we have $-M \leq g$, $-M \leq h$, and

$$V_{g,h}^\pi(x) = \max\left\{h(x), \min\left\{g(x), \gamma \cdot \mathbb{E}_{\pi,x'}\left[V_{g,h}^\pi(x')\right]\right\}\right\} \geq h(x) \geq -M \tag{27}$$

According to Lemma 4.4, the Bellman equation (27) is a contraction mapping and has a unique solution, denoted as $V_{g,h}^\pi(x)$.

**Step 1: For $x \in \mathcal{T}$**

$$V_{g,h}^\pi(x) \geq h(x) = -M$$

**Step 2: For $x \in \mathcal{F}$**

For $x \in \mathcal{F}$, we have $h(x) > 0, g(x) > 0$. By the definition of $V_{g,h}^\pi(x)$, we know that

$$V_{g,h}^\pi(x) \geq h(x) > 0$$

**Step 3: For $x \notin \mathcal{T} \cup \mathcal{F}$ with $V_{g,h}^\pi(x) < 0$**

By the definition of $V_{g,h}^\pi(x)$, $V_{g,h}^\pi(x) < 0$, $h(x) = M$ if $x \in \mathcal{F}$, $h(x) = -M$ otherwise, $g(x) > 0$ if $x \notin \mathcal{T}$, and $\gamma \in (0, 1)$, we have:

$$0 > \gamma \cdot \mathbb{E}_{\pi,x'}\left[V_{g,h}^\pi(x')\right] \geq \gamma h(x') \geq \gamma M > -M \Rightarrow \min\left\{g(x), \gamma \cdot \mathbb{E}_{\pi,x'}\left[V_{g,h}^\pi(x')\right]\right\} > h(x) \tag{28}$$

$$\Rightarrow \max\left\{h(x), \min\left\{g(x), \gamma \cdot \mathbb{E}_{\pi,x'}\left[V_{g,h}^\pi(x')\right]\right\}\right\} = \min\left\{g(x), \gamma \cdot \mathbb{E}_{\pi,x'}\left[V_{g,h}^\pi(x')\right]\right\} \tag{29}$$

If $V_{g,h}^\pi(x) = g(x)$, this conflicts with the conditions $V_{g,h}^\pi(x) < 0$ and $g(x) > 0$. Therefore, under these conditions, it must be that $V_{g,h}^\pi(x) = \gamma \cdot \mathbb{E}_{\pi,x'}\left[V_{g,h}^\pi(x')\right]$. $\square$

So, $\frac{-V_{g,h}^\pi(x)}{M}$ satisfies (25) on the set $\mathcal{X} \setminus \{x \mid V_{g,h}^\pi(x) > 0\}$. If $\frac{1}{M}V_{g,h}^\pi(x) < 0$, then the following equation holds:

$$\mathbb{P}_\pi(\mathbf{RA}_x) \geq -\frac{1}{M}V_{g,h}^\pi(x). \tag{30}$$

## B. Gradients Derivation and Algorithm Convergence

### B.1. Gradient estimates

For this stochastic value function (9), the Q function (Sutton et al., 1998) is defined as

$$Q_{g,h}^\pi(x, a) = \max\left\{h(x), \min\left\{g(x), \gamma \mathbb{E}_{x' \sim P(x'|x,a)}[V_{g,h}^\pi(x')]\right\}\right\}.$$

The Q value losses are based on the MSE between the Q networks and TD targets, gradients of which are shown below:

$$\hat{\nabla}_\eta J_Q(\eta) = \nabla_\eta Q_{g,h}(x_t, a_t) \cdot \left[Q_{g,h}(x_t, a_t) - \max\left\{h(x_t), \min\left\{g(x_t), \gamma Q_{g,h}(x_{t+1}, a_{t+1})\right\}\right\}\right] \tag{31}$$

$$\hat{\nabla}_\kappa J_{Q_c}(\kappa) = \nabla_\kappa Q_c(x_t, a_t) \cdot \left[Q_c(x_t, a_t) - (c(x_t, a_t) + \gamma Q_c(x_{t+1}, a_{t+1}))\right] \tag{32}$$

We show the gradient update of compensation factor is:

$$\hat{\nabla}_\xi J_\phi(\xi) = \nabla_\xi \phi_\xi(x_t) \cdot \left( \phi_\xi(x_t) - y_t \right). \tag{33}$$

then we compute the gradient with respect to $\theta$. First, we propose an auxiliary MDP(So et al., 2024) that facilitates the computation of gradients.

**Definition B.1** (Absorbing Markov Decision Process). The Absorbing Markov Decision Process is defined on f with an added absorbing state $s$. We define the transition function $f'_r$ with the absorbing state as

$$f'_r(x, a) = \begin{cases} f(x, a), & \text{if } g(x) \geq V^\pi_{g,h}(x) \geq h(x), \\ s, & \text{otherwise.} \end{cases} \tag{34}$$

Denote by $d'_\pi(x)$ the stationary distribution under stochastic policy $\pi$.

Now we give a policy gradient theorem for the absorbing MDP above.

**Theorem B.2.** *(Policy Gradient Theorem) For policy $\pi_\theta$ parameterized by $\theta$, the gradient of the policy value function $V^{\pi_\theta}_{g,h}$ satisfies*

$$\nabla_\theta V^{\pi_\theta}_{g,h}(x) \propto \mathbb{E}_{x' \sim d'_\pi(x), a \sim \pi_\theta} \left[ Q^{\pi_\theta}(x', a) \nabla_\theta \ln \pi_\theta(a \mid x') \right], \tag{35}$$

*under the stationary distribution $d'_\pi(x)$ for Reachability MDP in Definition B.1.*

*Proof.*

$$\begin{aligned}
\nabla_\theta V^{\pi_\theta}_{g,h}(x) &= \nabla_\theta \left( \sum_{a \in \mathcal{U}} \pi_\theta(a \mid x) Q^{\pi_\theta}_{g,h}(x, a) \right) \\
&= \sum_{a \in \mathcal{A}} \left( \nabla_\theta \pi_\theta(a \mid x) Q^{\pi_\theta}_{g,h}(x, a) + \pi_\theta(a \mid x) \nabla_\theta Q^{\pi_\theta}_{g,h}(x, a) \right) \\
&= \sum_{a \in \mathcal{A}} \left( \nabla_\theta \pi_\theta(a \mid x) Q^{\pi_\theta}_{g,h}(x, a) \right. \\
&\qquad \left. + \pi_\theta(a \mid x) \nabla_\theta \max\{h(x), \min\{g(x), \gamma V^\pi_{g,h}(x')\}\} \right) \\
&= \sum_{a \in \mathcal{A}} \left( \nabla_\theta \pi_\theta(a \mid x) Q^{\pi_\theta}_{g,h}(x, a) \right. \\
&\qquad \left. + \pi_\theta(a \mid x) 1_{x' \in \{x' \mid \max[h(x), \min[g(x), \gamma V^\pi_{g,h}(x')]] = V^{\pi_\theta}_{g,h}(x')\}} \gamma \nabla_\theta V^{\pi_\theta}_{g,h}(x') \right)
\end{aligned} \tag{36}$$

where $x' = f'_r(x, a)$

By unrolling $V^{\pi_\theta}_{g,h}(x')$ under extended MDP in Definition B.1 and defining $\mathbb{P}(x \to x^\star, k, \pi_\theta)$ as the probability of transitioning from state $x$ to $x^\star$ in $k$ steps under policy $\pi_\theta$ in B.1, we can get

$$\nabla_\theta V^{\pi_\theta}_{g,h}(x) = \sum_{x^\star \in \mathcal{X}} \left( \sum_{k=0}^\infty \mathbb{P}(x \to x^\star, k, \pi) \right) \sum_{a \in \mathcal{A}} \nabla \pi_\theta(a \mid x^\star) Q^{\pi_\theta}_{g,h}(x^\star, a) \tag{37}$$

$$\propto \mathbb{E}_{x' \sim d'_\pi(x), a \sim \pi_\theta} \left[ Q^{\pi_\theta}_{g,h}(x', a) \nabla_\theta \ln \pi_\theta(a \mid x') \right] \tag{38}$$

Note that $1_{x' \in \{x' \mid \max[h(x), \min[g(x), \gamma V^\pi_{g,h}(x')]] = V^{\pi_\theta}_{g,h}(x')\}}$ is absorbed using the absorbing state in Definition B.1.

$\square$

Similarly, we could have:

$$\nabla_\theta V_{g,h}^{\pi_\theta}(x_t) = \mathbb{E}_{x' \sim d'_\pi(x), a \sim \pi_\theta} \left[ \gamma^t Q^{\pi_\theta}(x', a) \nabla_\theta \ln \pi_\theta(a \mid x') \right], \tag{39}$$

under the stationary distribution $d'_\pi(x)$ for Reachability MDP in Definition B.1.

## B.2. Convergence Analysis

We provide convergence analysis of our algorithm for Finite MDPs. We demonstrate our algorithm almost surely finds a locally optimal policy for our RAPC-PPO formulation, based on the following assumptions:

- **A1 (Step size):** Step sizes follow schedules $\{\zeta_1(k)\}, \{\zeta_2(k)\}, \{\zeta_3(k)\}$ where:

$$\sum_k \zeta_i(k) = \infty \quad \text{and} \quad \sum_k \zeta_i(k)^2 < \infty, \forall i \in \{1, 2, 3\},$$

and

$$\zeta_j(k) = o(\zeta_{j-1}(k)), \forall j \in \{2, 3\}.$$

The cost returns critic value functions and RAPC must follow the fastest schedule $\zeta_1(k)$, the policy must follow the second fastest schedule $\zeta_2(k)$, the compensation factor must follow the second slowest schedule $\zeta_3(k)$.

- **A2 (Differentiability and Lipschitz Continuity):** For all state-action pairs $(s, a)$, we assume RAPC and cost Q functions $V_{g,h}(s; \eta)$, $V_c(s; \kappa)$, policy $\pi(a|s; \theta)$, and compensation factor $\phi(s; \xi)$ are continuously differentiable in $\eta$, $\kappa$, $\theta$, $\xi$ respectively. Furthermore, $\nabla_\theta \pi(a|s; \theta)$ is Lipschitz continuous functions in $\theta$.

- **A3 (Strict Feasibility):** $\exists \pi(\cdot|\cdot; \theta)$ such that $\forall x \in \mathcal{X}^-$ where $\phi_\gamma^\pi(x) > 0$, $V_{g,h}^\pi(s) \leq -\epsilon \cdot \phi_\gamma^\pi(x)$.

**Theorem B.3.** *Given Assumptions A1-A3, the policy updates in Algorithm 1 will almost surely converge to a locally optimal policy for our proposed optimization in Equation RAPC-PPO.*

*Proof.* The proof follows the standard multi-timescale stochastic approximation argument for actor–critic algorithms and closely parallels the analysis in (Ganai et al., 2023; Gu et al., 2024). We provide a brief proof sketch.

- *Critic convergence.* Under Assumptions 1-3, the RAPC critic and the cost critic are updated on a faster timescale than the policy parameters. For a fixed policy $\pi_\theta$, the corresponding Bellman-type operators are $\gamma$-contractions. Therefore, by standard stochastic approximation results, the critic parameters converge almost surely to their respective fixed points.

- *Policy convergence.* Due to the timescale separation, the critics can be treated as quasi-stationary when analyzing the policy update. The resulting policy recursion is a projected stochastic gradient descent with a gated (soft-switched) update direction. This update has the same structure as the policy optimization scheme in (Gu et al., 2024), for which it is shown that the policy iterates converge almost surely to a stationary point that is locally optimal. Hence, the policy parameters $\theta_k$ converge almost surely to a locally optimal policy.

This completes the proof. □

## C. On-Policy Details

### C.1. GAE estimator Definition

Note, however, that the definition of return (47) is *different* from the original definition and hence will result in a different equation for the GAE. The Bellman equation:

$$V_{g,h}^\pi(x_t) = \mathbb{E}_{a_t \sim \pi, x_{t+1} \sim P} \left[ \max\{h(x_t), \min\{g(x_t), \gamma V_{g,h}^\pi(x_{t+1})\}\} \right] \tag{40}$$

Define the $k$-step target $\Phi^{(k)}$ for the trajectory segment $x_t, \ldots, x_{t+k}$ using the operator $T(x, V_{\text{next}}) = \max\{h(x), \min\{g(x), \gamma V_{\text{next}}\}\})$. The target is built recursively, bootstrapping from the value function $V_{g,h}^\pi$ evaluated at the final state $x_{t+k}$:

$$\Phi^{(k)}(x_t, \ldots, x_{t+k}) = T(x_t, \Phi^{(k-1)}(x_{t+1}, \ldots, x_{t+k})) \quad \text{for } k \geq 1 \tag{41}$$

where the recursion implicitly uses $\Phi^{(0)}(x_{t+k}) = V_{g,h}^{\pi}(x_{t+k})$ as the base case when unwound. For instance:

$$\Phi^{(1)}(x_t, x_{t+1}) = T(x_t, V_{g,h}^{\pi}(x_{t+1}))$$

$$\Phi^{(2)}(x_t, x_{t+1}, x_{t+2}) = T(x_t, T(x_{t+1}, V_{g,h}^{\pi}(x_{t+2})))$$

The $k$-step advantage estimator $\hat{A}_{g,h}^{\pi(k)}$ depends on the trajectory segment:

$$\hat{A}_{g,h}^{\pi(k)}(x_t, \ldots, x_{t+k}) = \Phi^{(k)}(x_t, \ldots, x_{t+k}) - V_{g,h}^{\pi}(x_t) \tag{42}$$

The GAE estimator $\hat{A}_{g,h}^{\pi(\text{GAE})}$ is the $\lambda$-weighted sum:

$$\hat{A}_{g,h}^{\pi(\text{GAE})}(x_t) = \frac{1}{1-\lambda} \sum_{k=1}^{\infty} \lambda^k \hat{A}_{g,h}^{\pi(k)}(x_t, \ldots, x_{t+k}) \tag{43}$$

Note: The calculation of $\hat{A}_{g,h}^{\pi(k)}$ inside the sum requires evaluating $V_{g,h}^{\pi}$ and the functions $h, g$ along the trajectory segment $x_t, \ldots, x_{t+k}$.

### C.2. On-Policy RAPCPO Deployment Details

We denote the policy optimization step when the parameters are $\theta = \theta_l$ as follows:

$$\min_{\pi} \mathbb{E}_{x,a \sim \pi_{\theta_l}} \left[ \overline{A_c^{\pi_{\theta_l}}}(x, a) \mathbf{1}_{x \in \mathcal{X}_p^{\pi_{\theta_l}}} + \frac{\overline{A_{g,h}^{\pi_{\theta_l}}}(x, a)}{\phi_{\gamma}^{\pi_{\theta_l}}(x)} \mathbf{1}_{x \notin \mathcal{X}_p^{\pi_{\theta_l}}} \right] \tag{44}$$

$$\text{s.t.} \quad \forall x \in \mathcal{X}_p^{\pi_{\theta_l}}, \quad V_{g,h}^{\pi}(x) \leq -M \cdot \phi_{\gamma}^{\pi_{\theta_l}}(x) \cdot p.$$

$$\mathcal{X}_p^{\pi_{\theta_l}} = \left\{ x \in \mathcal{X} \ \middle| \ V_{g,h}^{\pi_{\theta_l}}(x) \leq -pM\phi_{\gamma}^{\pi_{\theta_l}}(x), \ \phi_{\gamma}^{\pi_{\theta_l}}(x) \geq 0 \right\}, \tag{45}$$

$$\overline{A_c^{\pi_{\theta_l}}}(x, a) = \max \left( \frac{\pi_\theta(a \mid x)}{\pi_{\theta_l}(a \mid x)} \hat{A}_c^{\pi_{\theta_l}(\text{GAE})}(x, a), \ \text{CLIP} \left( \frac{\pi_\theta(a \mid x)}{\pi_{\theta_l}(a \mid x)}, 1 - \epsilon, 1 + \epsilon \right) \hat{A}_c^{\pi_{\theta_l}(\text{GAE})}(x, a) \right) \tag{46}$$

$$\overline{A_{g,h}^{\pi_{\theta_l}}}(x, a) = \max \left( \frac{\pi_\theta(a \mid x)}{\pi_{\theta_l}(a \mid x)} \hat{A}_{g,h}^{\pi_{\theta_l}(\text{GAE})}(x, a), \ \text{CLIP} \left( \frac{\pi_\theta(a \mid x)}{\pi_{\theta_l}(a \mid x)}, 1 - \epsilon, 1 + \epsilon \right) \hat{A}_{g,h}^{\pi_{\theta_l}(\text{GAE})}(x, a) \right) \tag{47}$$

Let $r_t(\theta) = \pi_\theta(a_t \mid x_t)/\pi_{\theta_l}(a_t \mid x_t)$ denote the usual PPO probability ratio. Since we formulate policy optimization as minimizing a loss, we use the PPO clipped *loss* form. Define the reach-avoid and cost policy losses as

$$\ell_R(t; \theta) = \max \left( r_t(\theta) \overline{A_{g,h}^{\pi_\theta}}, \ \text{clip}(r_t(\theta), 1 - \epsilon, 1 + \epsilon) \overline{A_{g,h}^{\pi_\theta}} \right), \tag{48}$$

$$\ell_C(t; \theta) = \max \left( r_t(\theta) \overline{A_{g,h}^{\pi_\theta}}, \ \text{clip}(r_t(\theta), 1 - \epsilon, 1 + \epsilon) \overline{A_{g,h}^{\pi_\theta}} \right), \tag{49}$$

At each policy update iteration, we partition an on-policy minibatch $\mathcal{B}$ into two disjoint subsets according to the surrogate-feasible indicator $m(x_t)$:

$$\mathcal{B}_1 = \{(x_t, a_t) \in \mathcal{B} : m(x_t) = 1\}, \quad \mathcal{B}_0 = \{(x_t, a_t) \in \mathcal{B} : m(x_t) = 0\}, \tag{50}$$

which form a partition of $\mathcal{B}$. Samples in $\mathcal{B}_0$ are treated as surrogate-infeasible and are used exclusively to improve the reach-avoid surrogate objective, while samples in $\mathcal{B}_1$ are surrogate-feasible and are used to jointly optimize reach-avoid performance and cost.

Using the samples in $\mathcal{B}_1$, we compute the reach-avoid and cost policy gradients

$$g_R^{(1)} = \nabla_\theta \mathbb{E}_{(x_t, a_t) \in \mathcal{B}_1} \left[ \ell_R(t; \theta) \right] \Big|_{\theta = \theta_l}, \quad g_C^{(1)} = \nabla_\theta \mathbb{E}_{(x_t, a_t) \in \mathcal{B}_1} \left[ \ell_C(t; \theta) \right] \Big|_{\theta = \theta_l}. \tag{51}$$

When the two gradients are negatively aligned, $\langle g_R^{(1)}, g_C^{(1)} \rangle < 0$, we remove conflicting components via a symmetric projection step, inspired by projection-based gradient conflict resolution methods in multi-objective optimization:

$$\tilde{g}_R^{(1)} = g_R^{(1)} - \frac{\langle g_R^{(1)}, g_C^{(1)} \rangle}{\langle g_C^{(1)}, g_C^{(1)} \rangle + \delta} g_C^{(1)}, \tag{52}$$

$$\tilde{g}_C^{(1)} = g_C^{(1)} - \frac{\langle g_C^{(1)}, g_R^{(1)} \rangle}{\langle g_R^{(1)}, g_R^{(1)} \rangle + \delta} g_R^{(1)}, \tag{53}$$

where $\delta > 0$ is a small numerical stabilization constant. The mixed feasible-set update direction is then defined as

$$g_{\mathrm{mix}} = \begin{cases} \tilde{g}_R^{(1)} + \tilde{g}_C^{(1)}, & \langle g_R^{(1)}, g_C^{(1)} \rangle < 0, \\ g_R^{(1)} + g_C^{(1)}, & \text{otherwise.} \end{cases}$$

Using the surrogate-infeasible samples in $\mathcal{B}_0$, we compute the reach-avoid gradient

$$g_R^{(0)} = \nabla_\theta \, \mathbb{E}_{(x_t, a_t) \in \mathcal{B}_0} \left[ \ell_R(t; \theta) \right] \Big|_{\theta = \theta_l}.$$

The final policy update direction is obtained by combining the two components:

$$g = g_R^{(0)} + g_{\mathrm{mix}},$$

and the actor parameters are updated via

$$\theta_{l+1} = \theta_l - \eta \, g.$$

## D. Experiment Details

In this section, we provide additional experimental details, including the hardware setup, environment configurations, baseline implementations, and hyperparameter settings.

All experiments were conducted on a system equipped with a NVIDIA RTX4090 GPU. The training process for each environment took a maximum of 1 hours to complete.

### D.1. Environment Settings

Both the experimental setup of our main experiments and the reward shaping design for the baselines follow (So et al., 2024), with a notable distinction in the definition of the function $g$. Specifically, while (So et al., 2024) employs state augmentation, defining $g$ based on an augmented state $\tilde{x}$, the function $g(\tilde{x})$ in their work effectively depends only on the original state component $x$. Recognizing this, our method simplifies the formulation by defining $g$ directly on the original state $x$, thereby avoiding the need for state augmentation.

Apart from this change, the remaining experimental configurations for the Safety HalfCheetah, Safety Hopper, Pendulum, and FixedWing environments are kept consistent with those in (So et al., 2024), to ensure a fair comparison. For the PointGoal environment, the cost is defined as $c(x_t, a_t, x_{t+1}) = 3 \cdot \|a_t\|^2$. All six environments are implemented in Jax (Bradbury et al., 2018) to enable improved computational efficiency and parallelization.

### D.2. Supplementary Experiments

Here, we compare the true reach-avoid probability with estimates obtained using the fixed-$\gamma$ Bellman equation in (8), with and without the compensation factor in Figure 8.

To further clarify the behavior of the learned cost critic, we conducted additional experiments and report a fine-grained decomposition of average costs over safe, unsafe, and all trajectories. This analysis is intended to show that the cost critic preserves a complete view of the cost landscape, rather than only reflecting trajectories that eventually satisfy the safety constraint. For successful safe trajectories, the cost is accumulated until the completion time $T$, where typically $T < H$. For

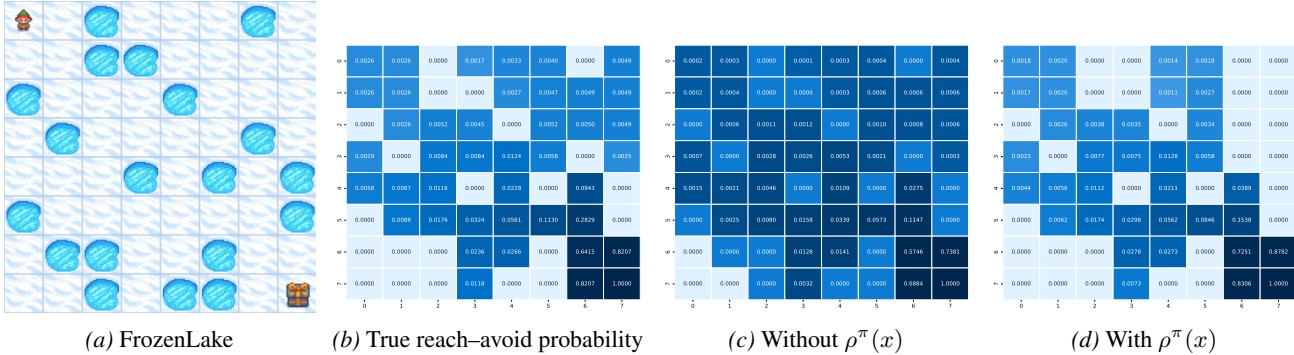

*(a)* FrozenLake     *(b)* True reach–avoid probability     *(c)* Without $\rho^\pi(x)$     *(d)* With $\rho^\pi(x)$

*Figure 8.* Effect of the auxiliary compensation factor on reach–avoid probability estimation. The compensation factor significantly reduces conservativeness.

*Table 3.* Average cost decomposition over safe, unsafe, and all trajectories. Lower values indicate lower accumulated cost.

| Task | Method | Safe | Unsafe | All |
|---|---|---|---|---|
| PointGoal | RC-PP | 100.77 | 131.77 | 108.34 |
| | RESPO | 124.51 | 165.07 | 132.57 |
| | PPO$_\beta$ | 224.44 | 265.82 | 236.19 |
| | CPPO | 82.77 | 109.25 | 97.42 |
| | Saute | 60.97 | 81.77 | 73.58 |
| | **Ours** | **58.23** | **74.07** | **61.53** |
| FixedWing | RC-PP | 54.09 | 74.09 | 56.72 |
| | RESPO | 75.83 | 100.25 | 77.83 |
| | PPO$_\beta$ | 192.16 | 251.35 | 202.64 |
| | CPPO | 43.27 | 57.93 | 51.36 |
| | Saute | 36.19 | 48.84 | 44.24 |
| | **Ours** | **35.52** | **46.86** | **36.82** |
| SafetyHopper | RC-PP | 63.82 | 80.29 | 65.18 |
| | RESPO | 72.27 | 96.04 | 73.41 |
| | PPO$_\beta$ | 102.32 | 131.25 | 106.53 |
| | CPPO | 59.03 | 79.32 | 68.79 |
| | Saute | 47.46 | 63.94 | 56.92 |
| | **Ours** | **42.13** | **53.29** | **43.17** |
| SafetyHalfCheetah | RC-PP | 66.37 | 88.43 | 71.45 |
| | RESPO | 78.33 | 102.73 | 82.26 |
| | PPO$_\beta$ | 111.34 | 152.66 | 121.87 |
| | CPPO | 46.89 | 63.23 | 57.17 |
| | Saute | 41.96 | 56.47 | 51.36 |
| | **Ours** | **40.33** | **50.72** | **42.65** |

unsuccessful unsafe trajectories, the cost is accumulated over the full horizon $H$. The resulting average costs are reported in Table 3.

As shown in Table 3, our method achieves the lowest average cost consistently across safe, unsafe, and all trajectories. We note that some baselines, such as Saute and CPPO, can also obtain relatively low costs. However, these results should be interpreted together with their reach rates reported in the main paper. In particular, these methods often reduce cost by failing to complete the task and remaining in safe but unproductive regions. In contrast, our method maintains low accumulated cost while achieving a higher reach rate, indicating that it learns to complete the task both safely and efficiently.

### D.3. Implementation of The Baselines

The implementation of the baseline follows their original implementations:

- **RC-PPO**: https://oswinso.xyz/rcppo/ (No license)

- **RESPO**: https://github.com/milanganai/milanganai.github.io/tree/main/NeurIPS2023/code (No license)

- **CPPO**: https://github.com/yingchengyang/CPPO (No license)

- **Sauté**: https://github.com/huawei-noah/HEBO/tree/master/SIMMER (No license)

### D.4. Hyperparameters

In experiments, the hyperparameters used by RAPC-PPO and other on-policy baselines are shown in Table 4.

*Table 4.* Hyperparameter Settings for On-policy Algorithms

| Hyperparameters for On-policy Algorithms | Values |
|---|---|
| **On-policy parameters** | |
| Network Architecture | MLP |
| Units per Hidden Layer | 256 |
| Numbers of Hidden Layers | 2 |
| Hidden Layer Activation Function | silu |
| Entropy coefficient | Linear Decay 1e-2 $\to$ 0 |
| Optimizer | Adam |
| Discount factor $\gamma$ | 0.99 |
| GAE lambda parameter | 0.95 |
| Clip Ratio | 0.2 |
| Actor Learning rate | Linear Decay 3e-4 $\to$ 0 |
| Reward/Cost Critic Learning rate | Linear Decay 3e-4 $\to$ 0 |
| **RAPC-PPO specific parameters** | |
| Discount factor $\gamma$ | 0.999 |
| Compensation factor Output Layer Activation Function | sigmoid |
| Compensation factor Learning Rate | Linear Decay 1e-4 $\to$ 0 |
| **RESPO specific parameters** | |
| REF Output Layer Activation Function | sigmoid |
| Lagrangian multiplier Output Layer Activation function | softplus |
| Lagrangian multiplier Learning rate | Linear Decay 5e-5 $\to$ 0 |
| REF Learning Rate | Linear Decay 1e-4 $\to$ 0 |

For static weighted multiplier $\beta$, we set $\beta = 0.1$. Moreover, the hard safety constraint $C_{fail}$ in (24) is set to 20 in every environment.

Regarding the difference in the discount factor $\gamma$ between RC-PPO and the other baselines, we note that the baselines were also evaluated with $\gamma = 0.999$; however, their performance was significantly worse than with $\gamma = 0.99$.

