# OpenReview forum: "Stochastic Minimum-Cost Reach-Avoid Reinforcement Learning"
_ICML.cc/2026/Conference — ICML 2026 regular_

### Official Review · Reviewer_GJ9s · 2026-02-21

**Soundness:** 3
**Presentation:** 4
**Significance:** 3
**Originality:** 3
**Overall Recommendation:** 5
**Confidence:** 4

**Summary:**

The paper considers RL tasks with minimum-cost reach-avoid specifications: the primary goal of the agent is to ensure that with probability $\geq$ a given threshold p, the target set of states T is reached while a set of failure states F is avoided. The secondary objective is to minimize the conditional expected cost along the runs that reach T and avoid F.

The approach builds on the existence of Lyapunov-like certificates for the primary reach-avoid objective developed within the formal methods community. The paper presents a new DP formulation for the certificate, which is then employed to design a critic estimating the probability of satisfying the primary constraint. This component is then coupled with a standard cost critic in an actor-critic framework where the actor is updated by a mixture of cost and reach-avoid gradients using symmetric projection-based gradient rectification.

An experimental evaluation is provided on a set of deterministic reach-avoid benchmarks and stochastic safety tasks from MuJoCo. The baselines consist of reach-avoid-specific deterministic approaches (RC-PPO) and generic constrained RL approaches. For the latter, the reach-avoid tasks are replaced with surrogate constraint MDP problems. The proposed approach, RAPCPO, demonstrates reach-avoid rates comparable with the better baselines (though slightly inferior to RESPO), while significantly improving on the expected cost. Ablations w.r.t. two components/hyperparameters are presented.

**Compliance With Llm Reviewing Policy:**

Affirmed.

**Final Justification:**

I thank the authors for the clarifying discussion. I continue to support the acceptance of the paper.

**Key Questions For Authors:**

[Q1] Please discuss the relationship to Lyapunov-based methods, see also the assessment above. E.g. in Chow et al., “A Lyapunov-based Approach to Safe Reinforcement Learning”, I understand that the certificates are recomputed on-line during the execution of the algorithm. Can you reasonably claim that your approach is the first to estimate Lyapunov-like certificates with critics obtained from a DP reformulation?

[Q2] How many seeds were used for each experiment?

[Q3] It is somewhat questionable whether ignoring the cost of trajectories that fail the reach-avoid spec should be ignored: what if the agent compensates for the improved cost on good trajectories by high-cost behavior on the failed ones? Do you have experimental data to address this? Wouldn’t it make sense to consider the cost over all trajectories? (Which should be well-defined with discounting).

[Q4] It is an interesting observation that RAPCPO achieves very good reach-avoid performance when the threshold is set to a relatively low constant, but the training fails if we set it to high values which are nonetheless achieved by the agent who trains with the low constant. This might present some challenges in practical deployment, where we would need to employ artificially low constants and hope that the agent trains to behave more “conservatively” than spec’ed. How would you recommend to address such an issue?

**Limitations:**

The limitations are discussed adequately.

**Strengths And Weaknesses:**

**Significance:**

+The problem addressed is relevant for safe RL. It is good that the paper considers constraints beyond plain accumulated costs.

+The presented method for estimating the certificates using the DP reformulation might be of independent interest for the safe RL community.

-The conditional expectation semantics, where the cost of paths that do not satisfy the reach-avoid spec is ignored, is questionable from a practical perspective.

**Presentation:**

+The paper is very clearly written and easy to follow.

**Soundness:**

+While I did not check all the proofs in detail, the theoretical developments follow a logical and clearly fleshed-out approach.

-One might argue that the number of environments in the experimental evaluation is somewhat limited (5), though this is to some degree justified by the lack of reach-avoid specific baselines. Also, the ablation studies seems to miss indicators of variance/confidence bounds.

**Originality:**

+The paper provides a nice example of a “technology transfer” from formal methods to RL and provides non-trivial scaffolding to make such a transfer possible.

-What could be discussed better is the relationship between the proposed approach to Lyapunov-based methods in safe RL. I believe that the claim in Sec 2 that “Lyapunov based methods rely on hand-crafted certificates” might not be entirely accurate, see also my questions below.

---

> ### Author Rebuttal · Authors · 2026-03-31
>
> We sincerely thank the reviewer for the thorough and constructive evaluation. Below we address each point in detail.
>
> # Q1 (Relationship with Lyapunov-based methods):
> We agree that the statement in Section 2, "However, they rely on hand-crafted certificates," was too brief and may have given an overly narrow impression. In the revised manuscript, we have expanded this discussion to provide a more balanced overview of Lyapunov-based safe RL methods, including works such as [1], where the certificate may be updated during the algorithm rather than fixed a priori. We also clarify that we do not claim to be the first to estimate Lyapunov-like certificates via critic learning or DP reformulations. For instance, Bellman-based approaches for estimating reach–avoid quantities have been studied in prior work [2].
>
> Our contribution instead lies in the problem formulation and the structure of the resulting certificate. Most Lyapunov-based methods enforce safety by guaranteeing avoidance of unsafe states through per-state recursive constraints. In contrast, our framework targets probabilistic reach-avoid specifications within a stochastic minimum-cost setting. To support this, we introduce structured, non-sparse learning signals within a specialized DP formulation (Section 4.2), which yields a value function that serves as a lower bound on the reach–avoid probability under the given policy, following from the construction of the operator and the shaping functions g and  h. In optimization, this certificate is incorporated via a surrogate objective, which enables gradient-based policy updates while approximately enforcing the reach-avoid specification despite the sparsity of the underlying logical objective.
>
> *[1] Chow, Yinlam, et al. "A lyapunov-based approach to safe reinforcement learning." NIPS, 2018.*
>
> *[2] Xue, Bai. "Sufficient and necessary barrier-like conditions for safety and reach-avoid verification of stochastic discrete-time systems." Automatica (2026).*
>
> # Q2 & Weaknesses (Seeds, Variance, and Environments):
>
> While 5 environments is limited, we selected diverse tasks due to a lack of standard reach-avoid benchmarks. All results average 10 random seeds. To explicitly show robustness, we will add standard deviation error bars to all plots in the revision.
> # Q3 (Whether ignoring costs on failed trajectories is reasonable):
>
> We agree that, in principle, one could optimize cost over all trajectories, including those that fail the reach-avoid specification. Our method partially follows this approach, but deliberately introduces a hierarchical prioritization mechanism to ensure safety.
>
> Specifically, the expected cumulative cost is captured by the value function $V_c^\pi(x)$, which is learned via temporal-difference updates using transitions from all rollouts (Eq.~32). As a result, high-cost behavior on failed trajectories is fully reflected in the learned cost critic. However, during policy optimization, for states that violate the reach-avoid constraint, the update temporarily suppresses the cost gradient to prioritize feasibility. This reflects the intended hierarchy of the objective: when the specification is violated, the primary goal is to recover feasibility rather than optimize cost. Once feasibility is restored, cost minimization over all trajectories resumes. Therefore, while cost information from failed trajectories is fully learned and retained, it is intentionally not optimized against in infeasible regions, in order to avoid trade-offs that could compromise reach-avoid satisfaction.
>
> To directly address your question with experimental data, we add the following additional experiments. In the Safetyhopper environment, the average cost on successful trajectories is 42.13, while the average cost on failed trajectories is 53.29. This data confirms that the agent does not exploit failed rollouts to incur arbitrarily high costs.
>
> # Q4 (p of practical deployment):
>
> Training instability at large initial p arises because optimization becomes feasibility-dominated: much of the state space is treated as infeasible, so updates focus on safety recovery rather than exploration and cost reduction. In practice, we recommend starting with a moderate threshold (e.g., p=0.5), increasing it if reach-avoid success is insufficient and decreasing it if cumulative cost is too high.
> More generally, p should be viewed as a certified safety floor rather than an exact empirical target: because RAPCs provide a sufficient and typically conservative lower bound on reach-avoid probability (Theorem 4.5), policies trained with moderate thresholds often achieve higher empirical success rates.

---

> > ### Author Rebuttal · Reviewer_GJ9s · 2026-04-02
> >
> > I thank the authors for their detailed feedback. Regarding point 3, I suggest that the main text is updated with a comment pointing out that the cost critic retains the information from all rollouts. I also have a follow-up question regarding 3: could you present, based on your data, the comparison of avg costs over safe/unsafe/all eval trajectories across all environments and baselines? Ideally, in a form of a table.

---

> > > ### Author Response · Authors · 2026-04-06
> > >
> > > Thank you for your follow-up comments and the constructive suggestion. We have addressed your points in the revised manuscript to provide a more comprehensive analysis.
> > >
> > > 1. Regarding the Cost Critic:
> > > As suggested, we have updated the main text (Section 5.1) to explicitly clarify that the cost critic retains information from all rollouts. This ensures that the critic maintains a complete representation of the cost landscape, regardless of whether a specific trajectory was safe or unsafe.
> > >
> > > 2. Comparison of Average Costs (Safe vs. Unsafe Trajectories):
> > > Since our previous experiments did not separately report the costs of safe and unsafe trajectories, we have re-run the evaluations to provide a more granular view of performance, breaking down the average cost into Safe (successfully completed the reach-avoid task), Unsafe (failed), and All trajectories.
> > >
> > > For successful (safe) trajectories, costs are accumulated up to the time step T at which the task is completed (typically T < H). For unsuccessful (unsafe) trajectories, costs are accumulated over the full horizon H. The updated results are as follows:
> > >
> > > **PointGoal**
> > > | Method | Safe | Unsafe | All |
> > > |:---|:---:|:---:|:---:|
> > > | RC-PPO | 100.77 | 131.77 | 108.34 |
> > > | RESPO | 124.51 | 165.07 | 132.57 |
> > > | PPO_β | 224.44 | 265.82 | 236.19 |
> > > | CPPO | 82.77 | 109.25 | 97.42 |
> > > | Saute | 60.97 | 81.77 | 73.58 |
> > > | **Ours** | 58.23 | 74.07 | 61.53 |
> > >
> > > **FixedWing**
> > > | Method | Safe | Unsafe | All |
> > > |:---|:---:|:---:|:---:|
> > > | RC-PPO | 54.09 | 74.09 | 56.72 |
> > > | RESPO | 75.83 | 100.25 | 77.83 |
> > > | PPO_β | 192.16 | 251.35 | 202.64 |
> > > | CPPO | 43.27 | 57.93 | 51.36 |
> > > | Saute | 36.19 | 48.84 | 44.24 |
> > > | **Ours** | 35.52 | 46.86 | 36.82 |
> > >
> > > **SafetyHopper**
> > > | Method | Safe | Unsafe | All |
> > > |:---|:---:|:---:|:---:|
> > > | RC-PPO | 63.82 | 80.29 | 65.18 |
> > > | RESPO | 72.27 | 96.04 | 73.41 |
> > > | PPO_β | 102.32 | 131.25 | 106.53 |
> > > | CPPO | 59.03 | 79.32 | 68.79 |
> > > | Saute | 47.46 | 63.94 | 56.92 |
> > > | **Ours** | 42.13 | 53.29 | 43.17 |
> > >
> > > **SafetyHalfCheetah**
> > > | Method | Safe | Unsafe | All |
> > > |:---|:---:|:---:|:---:|
> > > | RC-PPO | 66.37 | 88.43 | 71.45 |
> > > | RESPO | 78.33 | 102.73 | 82.26 |
> > > | PPO_β | 111.34 | 152.66 | 121.87 |
> > > | CPPO | 46.89 | 63.23 | 57.17 |
> > > | Saute | 41.96 | 56.47 | 51.36 |
> > > | **Ours** | 40.33 | 50.72 | 42.65 |
> > >
> > > Discussion:
> > > We note that while baselines like Saute and CPPO achieve relatively low costs, this must be interpreted alongside their reach rates (approx. 0.4, as reported in Figure 2 and Figure 5). These methods often achieve low cost by failing to prioritize reachability, thus staying in "safe" but unproductive regions. In contrast, our method achieves the lowest cost across all categories while maintaining a significantly higher reach rate, demonstrating that it learns to complete the task both safely and efficiently.
> > >
> > > We have incorporated these tables and the related discussion into Appendix D of the revised manuscript.

---

### Official Review · Reviewer_rmMt · 2026-03-11

**Soundness:** 4
**Presentation:** 3
**Significance:** 4
**Originality:** 3
**Overall Recommendation:** 5
**Confidence:** 4

**Summary:**

Authors tackle an important problem of minimum cost reach-avoid reinforcement learning.
In MDP, the agent needs to satisfy a specification while minimizing the cost.
The paper introduces certificates of reach-avoid probabilities and costs and describe an algorithm that almost surely converges to optimal policies.
Also, the work examines the practical feasibility by empirical simulations against baselines.

**Compliance With Llm Reviewing Policy:**

Affirmed.

**Key Questions For Authors:**

No questions

**Limitations:**

Limitations section is very good.

**Strengths And Weaknesses:**

Soundness: Yes, the proofs are clear and, considering space constraint,s well-explained already in the main text.
 The methods and assumptions are appropriate.
The experimental results are well-designed.
Personally, I think that too much space was dedicated to the experiments, while the proofs are the most interesting part of the paper.
(but there is no need to change anything)
The limitations section is good.

Presentation: The submission is clearly written.
The structure is also good.
The narrative is easy to follow.
The work positions itself decently within the existing literature; learning with LTL specifications is also very relevant here.
LTL specifications contain reach-avoid specifications and many more.
For recent works on LTLs, Le et al. Neurips'24 and Svoboda et al. ICML'24 seems relevant here.

Significance: The problem is very relevant in theory and it advances machine learning.
Other researchers can build on ideas in the paper.
The scope of impact is mainly theoretical, but they can unlock new more practical directions.

Originality: The work provides new insights and introduces new techniques.
The novelty is well justified.

---

> ### Author Rebuttal · Authors · 2026-03-31
>
> We sincerely thank the reviewer for the positive evaluation, the encouraging assessment of the soundness and significance of our work, and the insightful suggestion regarding related work on LTL-based learning. We agree that LTL specifications, which subsume reach-avoid as a special case, represent an important and closely related direction. We have included a discussion of [1] and [2] in the revised manuscript to better position our work within the broader literature on temporal logic specifications in reinforcement learning.
>
> We also appreciate the opportunity to clarify the relation between our setting and the LTL-based literature. While our work is closely related in spirit, it focuses on a somewhat different objective: beyond enforcing a reach-avoid requirement, we study how to minimize cumulative cost while guaranteeing that the reach-avoid probability exceeds a prescribed threshold. Our certificates and optimization framework are specifically tailored to this cost-constrained setting, enabling principled trade-offs between performance and safety. In this sense, we view our formulation as complementary to prior LTL and reachability-based RL approaches, which primarily emphasize specification satisfaction. We believe this adds a useful perspective to the broader literature, and we have clarified this distinction in the text.
>
> *[1] Le, Xuan-Bach, et al. "Reinforcement learning with LTL and $\omega $-regular objectives via optimality-preserving translation to average rewards." Advances in Neural Information Processing Systems 37 (2024): 117109-117132.*
>
> *[2] Svoboda, Jakub, Suguman Bansal, and Krishnendu Chatterjee. "Reinforcement learning from reachability specifications: Pac guarantees with expected conditional distance." Forty-first International Conference on Machine Learning. 2024.*

---

> > ### Author Rebuttal · Reviewer_rmMt · 2026-04-02
> >
> > Thank you for the response and for the good work overall.
> > I agree that the objective of enforcing reach-avoid is a bit different, but I see the value in discussing the connection for the future work.

---

> > > ### Author Response · Authors · 2026-04-06
> > >
> > > Thank you for your positive feedback and for acknowledging our response. We appreciate your recognition of our work and your constructive suggestion regarding the connection between enforcing reach-avoid and related topics. Following your advice, we have incorporated a discussion on this connection in the revised manuscript to inspire future work.

---

### Official Review · Reviewer_6bKP · 2026-03-12

**Soundness:** 2
**Presentation:** 2
**Significance:** 3
**Originality:** 3
**Overall Recommendation:** 4
**Confidence:** 3

**Summary:**

The paper deals with a reach-avoid problem in the stochastic setting. It has been shown in the past that the standard safe RL formulation is sometimes not suited for the reach-avoid problem. Furthermore, the problem is met in the real-life setting very often. This necessiates further reseach and this paper is offering a step in this direction.

The authors derived a stochastic formulation for the reach-avoid problem, where the used the formalism of a hitting time to describe reaching the target set while avoiding the unsafe set. The formulation is flexible as it has a parameter $p$ indicating the probability of hitting the target set while avoiding the unsafe set. They derived a Bellman-like equation for it. They showed that the Bellman operator admits a unique fixed point.

Besides a theoretical contribution the authors derived a pratical algorithm based on PPO. Their formulation doesn't have a fixed time horizon and thefore additional care is needed for high hitting times. The authors derive a compensation factor $\phi$, which should help learning rate. They authors conducted a series of experiments. In particular, the answered the following questions

```
* Can RAPCPO achieve lower cost while maintaining a high reach-avoid probability in both deterministic and
stochastic tasks?
* How critical is the compensation factor $\phi$ to the performance of RAPCPO, as demonstrated through ablation studies?
* How does the parameter p influence the trade-off between cost and reach-avoid probability?
```

**Compliance With Llm Reviewing Policy:**

Affirmed.

**Key Questions For Authors:**

1. Line 136. Def 2. Wouldn’t a measure of the intersection of the reach tube and F be more appropriate in this definition?
1. I reiterate the question of formulation (6) from weaknesses. Can we be sure that the constraint is feasible?
1. I reiterate the question from weaknesses Lemma 4.4. Are there any conditions on g and h for the operator to admit the fixed point? Say if we can’t solve the reach-avoid problem with any p, then what will be the value function?
1. Eq 16. If $x \in X_p$ then the constraint is satisfied by construction - no need to add it. In fact, the constraint is not really playing a role in this formulation
1. Do we estimate $\phi$ only for the successful terminations? How do we estimate it for unsuccessful ones? I.e., if a point $x$ doesn’t have any rollout reaching $T$? Aren’t we supposed to maximize the second term for these kind of points?
1. Do we know the functions g and h? If we have an oracle is it part of problem formulation? In line 10 of the Algorithm does g refer to the shaped function? If so, why do we need to compute it?
1.  Mentioning that the algorithm is based on PPO more prominently will strengthen the paper. It's a strong base.
1.  I reiterate the question from weaknesses. The Saute algorithm assumes the almost surely setting which is equivalent to p=1, and the CVaR constraint also has a tunable parameter. It would be great to compare RAPCPO with p=1 with Saute and find a comparable percentile in the CVaR constraint with RACPO $p$ parameter. This comparison will be cleaner. At least a comment on this is needed.
1. Have the authors tried to schedule p in the ablation study? Say we pre-train a policy for p=0.7 and then start training with p=0.8?
1. Line 597. Proposition 1. Wording implies that the conditions are valid only for function $v$, but the conditions depend on  $\gamma$ too

**Limitations:**

Limitations are properly discussed.

**Strengths And Weaknesses:**

## Strengths:

* Novel formulation of the stochastic reach-avoid problem.
* Nice theoretical results, although I haven't checked the proofs.
* Experiments showing the advantages of the new algorithm
* An ablation study

## Weaknesses

1. Some of the theoretical statements are not entirely clear to me.
     * I am not sure that the formulation (6) is mathematically solid. The constraint $\forall x \in X_p: P_\pi(RA_x) \ge p$ seems to be very strict. This is because  the set $X_p$ is such that for any $x$ there exists a policy $\pi$ s.t $P_\pi(RA_x) \ge p$. But it seems that for different $x$ there could be different $\pi$. The existence of a policy $\pi$ satisfying the constraint for all $x$ is not obvious.
     * Lemma 4.4. Are there any conditions on g and h for the operator to admit the fixed point? Say if we can’t solve the reach-avoid problem with any p, then what will be the value function?
2. Algorithm presentation is slightly convoluted.
    * I had to dig deep to understand how the policy update is made. The fact that it's a PPO-type update was a pleasant surprise.
    * I am not entirely clear if the shaped functions g and h are known, learned or we have an oracle.
3. Some critic of experiments
    * The Saute algorithm assumes the almost surely setting which is equivalent to p=1, and the CVaR constraint also has a tunable parameter. It would be great to compare RAPCPO with p=1 with Saute and find a comparable percentile in the CVaR constraint with RACPO $p$ parameter. This comparison will be cleaner. At least a comment on this is needed.
    * The presence of PID Lagrangian algorithm can strengthen the paper (see https://github.com/PKU-Alignment/omnisafe/tree/main  or https://arxiv.org/abs/2007.03964)
    * The authors present a stochastic framework, but all the environments seem to be deterministic. It would be good to find a stochastic environment for a demonstration.

---

> ### Author Rebuttal · Authors · 2026-03-31
>
> We thank the reviewer for the insightful feedback.
>
> # W1/Q2/Q3: Mathematical rigor of Eq. (6) & Lemma 4.4
> Eq. (6) is solid because $\pi$ is inherently state-dependent (mapping states to actions), dynamically adapting to any state $x \in \mathcal{X}_p$, not a fixed sequence. Thus, a single policy can satisfy the constraint for all $x \in \mathcal{X}_p$ simultaneously, making Eq. (6) feasible if the set is non-empty. If $X_p^\pi$ becomes empty, it falls back to solely improving the reach-avoid probability (Eq. (6), second branch). Lemma 4.4 requires only bounded $g, h$ and $\gamma \in (0,1)$. Under these, $\mathcal{B}^\pi$ is a $\gamma$-contraction (under $|\cdot|\_\infty$), holding independently of $g, h$'s forms due to non-expansive $\max/\min$ operators. If the problem is unsolvable for any given $p$, $V^\pi\_{g,h}$ still exists but will not satisfy $-(1/M)V^\pi\_{g,h}(x) \ge p$ for any $x$, so $X^{\pi\_{\theta\_l}}\_p$ (Eq. (16)) becomes empty and the algorithm focuses entirely on improving reach-avoid probability (the second branch in Eq. (16)).
>
> # W2/Q6/Q7: Algorithm presentation & functions g, h
> We agree and have made the PPO basis more explicit in the revision.
> The functions $g$ and $h$ are known, task-specific signals (Eq. 1) encoding target-reaching and obstacle-avoidance. Like RL reward/cost functions, they are environment feedback, not learned or from an oracle. In Alg. 1 Line 10, $g$ denotes the policy gradient direction (Eqs. 20-22) not the shaping function $g(x)$. We will revise the notation to $g_\theta$.
>
> # W3/Q8: Baselines (Sauté/CVaR at p=1, PID Lagrangian, stochasticity)
> RAPCPO with $p=1$ on Safety Hopper yields Reach rate 0.8848 and cost 176. The core limitation of Sauté (Reach rate 0.4318) and CPPO-CVaR (Reach rate 0.5237) is that they **fail to achieve high reach-avoid rates**, confirming the structural misalignment between CMDP surrogates and the minimum-cost reach-avoid objective. Matching CVaR quantiles cannot recover our semantics of first ensuring $\mathbb{P}\_{\pi}(\mathbf{RA}\_{x\_0})\ge p$ then minimizing cost. We have clarified in Sec 6.
>
> **PID Lagrangian:** Following [1], we added PID Lagrangian (mean of 10 seeds):
>
> | Env | Method | Reach Rate | Cost |
> |-----|--------|------------|------|
> | SafetyHopper | RAPCPO | 0.9326 | 45.17 |
> | SafetyHopper | PID-Lag | 0.8949 | 121.35 |
> | SafetyHalfCheetah | RAPCPO | 0.7996 | 46.71 |
> | SafetyHalfCheetah | PID-Lag | 0.7283 | 108.76 |
> | PointGoal | RAPCPO | 0.7892 | 62.83 |
> | PointGoal | PID-Lag | 0.8297 | 132.93 |
> | FixedWing | RAPCPO | 0.9114 | 37.46 |
> | FixedWing | PID-Lag | 0.8897 | 217.98 |
>
> RAPCPO achieves significantly lower costs with comparable Reach rates. In PointGoal, PID-Lag achieves slightly higher Reach rate but at more than twice the cost.
>
> **Stochasticity:** SafetyHopper/SafetyHalfCheetah feature stochasticity via 10% Gaussian action noise per step (Sec 6).
>
> *[1] Stooke et al. "Responsive safety in RL by PID lagrangian methods." ICML, 2020.*
>
> # Q1 (Def. 2):
> Definition 3.1 defines $\text{RA}\_x$ as reaching $T$ while $\{x\_t\}\_{t=0}^{T(\tau)} \cap F = \emptyset$, which already captures the reach tube intersection with $F$.
>
> # Q4 (Eq. (16) constraint redundancy):
> Eq. (16) is not redundant: it governs the policy update toward the next iterate $\pi_\theta$, not evaluating the current one. It defines the feasible search space. For $x \in \mathcal{X}\_p^{\pi_{\theta_l}}$, it ensures the update focuses on cost minimization while ensuring that feasibility is preserved, i.e., preventing the reach-avoid probability from dropping below $p$. Without it, cost minimization alone could drive the policy into infeasible regions, leading to degradation of the reach-avoid specification.
>
> # Q5 (Estimation of $\phi$ for failures):
> Empirical targets $y_t = \gamma^{T-t}$ use only successful rollouts. Unsuccessful states rely on $\phi\_\gamma(x;\xi)$ generalizing from successful neighbors. To handle extrapolation bias and numerical instability, the denominator is bounded by $10^{-6}$ (Eq. 20). For states lacking successful rollouts ($V^\pi\_{g,h}>0$ and $x \notin \mathcal{X}\_p^{\pi\_{\theta\_l}}$), the cost objective drops, and only improve reach-avoid probability (the second branch in Eq. (16)).
>
> # Q9 (p-scheduling):
> SafetyHopper: pre-train $p=0.7 \to$ 0.934 Reach, 56.12 cost; then train $p=0.8 \to$ 0.951 Reach, 114.59 cost. Cost increases because $V\_{g,h}$ is a conservative lower bound struggling to satisfy Eq. (15) under noise for high $p$, shrinking $X^{\pi\_{\theta\_l}}\_p$ and reducing cost optimization. This reflects inherent RA guarantee vs. cost tension.
>
> # Q10 (Prop. 1):
> The reviewer is correct, the conditions implicitly depend on $\pi$. We will make this dependence explicit. Thank you.

---

> > ### Author Rebuttal · Reviewer_6bKP · 2026-04-02
> >
> > Thank you for the answers! I am mostly satisfied with the responses, but we need to sort out the maths here. It's possible that I am misunderstanding the formulation (6). I don't think it has an impact on the algorithm though.
> >
> > The way I read the equation (6) below:
> > $$\min_{\pi} E_{x,a \sim d^\pi_\rho} (V_c^\pi I_{X_p(x)} - P_\pi(RA_x) I_{X\setminus X_p(x) }) $$
> > $$\text{s.t.} \forall x \in X_p,  P_\pi(RA_x) \ge p$$
> >
> > is that $P_\pi(RA_x) \ge p$ has to satisfied for all $x$ in the set $X_p$ for our policy $\pi$ that we optimize over.
> >
> > The set $X_p$ is such that for every $x$ there exists a policy  $\tilde\pi$ such that $ P_{\tilde\pi}(RA_x) \ge p$. For two different $x$ two different policy can certify that $x\in X_p$.
> >
> > In my understanding $X_p$ is generally strictly bigger than $X_p^{\pi}$ and there exists an $\tilde x$ such that   $ P_{\pi}(RA_{\tilde x}) \not\ge p$ for many policies $\pi$. Therefore, I am not clear how there could be a policy that
> > $$\forall x \in X_p,  P_\pi(RA_x) \ge p$$
> > are we looking for policies where $X_p^\pi=X_p$? If so, is it easy to show?

---

> > > ### Author Response · Authors · 2026-04-06
> > >
> > > Thank you for your careful reading and follow-up question regarding the formulation of Eq. (6). We appreciate the opportunity to clarify the theoretical grounding of this definition.
> > >
> > > Yes, we are indeed looking for a setting where $X_p^{\pi^{\ast}} = X_p$ under an optimal policy $\pi^{\ast}$. To answer your question of whether this is easy to show: **yes, this follows directly from classical MDP theory.**
> > >
> > > **1. Global Optimality in Finite MDPs:**
> > >
> > > To build intuition, consider the standard finite-state, finite-action MDP setting. The optimal reach-avoid value function $V^{\ast}(x) = \sup_{\pi} V_{\pi}(x)$ satisfies the undiscounted Bellman optimality equation with absorbing states for target and failure sets:
> > >
> > > $$V^{\ast}(x) = \begin{cases} 1, & x \in \mathcal{T}, \\\\ 0, & x \in \mathcal{F}, \\\\ \max_{a \in \mathcal{A}} \sum_{x'} T(x' \mid x, a) \, V^{\ast}(x'), & \text{otherwise}. \end{cases}$$
> > >
> > > where $T(x' \mid x, a)$ denotes the transition probability from state x to state x′under action a. As established in classical MDP theory (e.g., Puterman [2], Bertsekas [3]), there always exists a **stationary deterministic policy** $\pi^{\ast}$ that is greedy with respect to $V^{\ast}$. Crucially, this single policy $\pi^{\ast}$ achieves $V_{\pi^{\ast}}(x) = V^{\ast}(x)$ for **every** state $x \in \mathcal{X}$ simultaneously.
> > >
> > > **2. Equivalence of $X_p^{\pi^{\ast}}$ and $X_p$:**
> > > - By definition, $x \in X_p$ if and only if $V^{\ast}(x) \geq p$.
> > > - Since $\pi^{\ast}$ achieves $V^{\ast}$ everywhere, for any $x \in X_p$, we have $P_{\pi^{\ast}}(RA_x) = V^{\ast}(x) \geq p$.
> > > - Thus, $x \in X_p \implies x \in X_p^{\pi^{\ast}}$, and obviously $X_p^{\pi^{\ast}} \subseteq X_p$,  which means $X_p^{\pi^{\ast}} = X_p$.
> > >
> > > Consequently, the optimization in Eq. (6) is well-posed: the optimal policy $\pi^{\ast}$ is a feasible solution that satisfies $P_{\pi^{\ast}}(RA_x) \geq p$ for all $x \in X_p$ simultaneously.
> > >
> > > **3. Extension to General/Continuous Settings:**
> > > While the exact uniform equivalence $X\_p^{\pi^\*} = X\_p$ is most easily shown in finite/discrete MDPs, similar results extend to continuous state-action spaces (which align with our experiments). For example, (Summers and Lygeros [1]) show that, in the finite-horizon continuous case, an optimal Markov policy exists under suitable compactness conditions.
> > >
> > > For the infinite-horizon case, their result [1] successfully characterizes the optimal value function through the Bellman equation, while the stationary deterministic optimal policy is stated only if it exists, indicating that additional assumptions are needed for policy existence. Indeed, guaranteeing a strictly optimal policy requires explicit topological assumptions (e.g., compactness and continuity), as is standard in measure-theoretic MDPs.
> > >
> > > However, even in the absence of such assumptions where a strict $\pi^\*$ might not be attainable, the foundational theory of MDPs on Borel spaces---specifically Positive Dynamic Programming (Bertsekas and Shreve [4], Chapter 5)---guarantees the existence of a uniformly $\epsilon$-optimal policy. Specifically, for any $\epsilon > 0$, there exists a policy $\pi_\epsilon$ such that
> > > $$
> > > P\_{\pi\_\epsilon}(R A\_x) \ge \sup\_{\pi} P\_{\pi}(R A\_x) - \epsilon
> > > \quad \text{for all } x \in \mathcal{X}.
> > > $$
> > >
> > > Because Eq.~(6) serves as the theoretical target formulation motivating our practical sampling-based algorithm, this foundational equivalence (or $\epsilon$-equivalence) ensures that the problem formulation is mathematically sound and well-posed, even in general continuous domains.
> > >
> > > ---
> > > **References:**
> > >
> > > [1] Summers, S., & Lygeros, J. (2010). Verification of discrete time stochastic hybrid systems: A stochastic reach-avoid decision problem. Automatica, 46(12), 1951-1961.
> > >
> > > [2] Puterman, M. L. (2014). Markov decision processes: discrete stochastic dynamic programming. John Wiley & Sons.
> > >
> > > [3] Bertsekas, D. (2012). Dynamic programming and optimal control: Volume I (Vol. 4). Athena scientific.
> > >
> > > [4] Bertsekas, D., & Shreve, S. E. (1996). Stochastic optimal control: the discrete-time case (Vol. 5). Athena Scientific.

---

### Official Review · Reviewer_E2mL · 2026-03-12

**Soundness:** 3
**Presentation:** 3
**Significance:** 3
**Originality:** 3
**Overall Recommendation:** 3
**Confidence:** 4

**Summary:**

This paper addresses an interesting problem relevant to the stochastic systems and reinforcement learning communities. The work studies stochastic minimum-cost reach–avoid reinforcement learning, where a learning agent must satisfy a reach–avoid specification while maintaining a computable lower bound on the probability of satisfaction. In addition to satisfying the reach–avoid objective, the agent is also tasked with minimizing the expected cumulative cost in a stochastic environment. The paper introduces reach–avoid probability certificates (RAPCs), which characterize the states from which stochastic reach–avoid constraints are satisfiable. These certificates are used to enforce the reach–avoid specification while simultaneously enabling cost optimization. The paper further establishes almost-sure convergence of the proposed algorithms to locally optimal policies under the reach–avoid constraints. Experimental results in the MuJoCo simulator demonstrate improved cost performance and higher reach–avoid satisfaction rates compared to existing baselines.

**Compliance With Llm Reviewing Policy:**

Affirmed.

**Key Questions For Authors:**

The proposed reach–avoid probability certificates are described as a principled mechanism for provably enforcing probabilistic reach–avoid constraints. Could the authors clarify whether these certificates provide a formal guarantee of the reach–avoid specification, and under what assumptions this guarantee holds?
Additionally, it is not entirely clear whether RAPCs admit a closed-form representation, or how they are constructed or synthesized in practice. Providing more details on the synthesis procedure would help clarify their practical applicability.
 Algorithm 1 is shown to converge to a locally optimal policy. Could the authors comment on the implications of this local optimality with respect to the claimed guarantees of reach–avoid constraint enforcement? In particular, does the lack of global optimality affect the robustness or reliability of the reach–avoid guarantees?
The RC-PPO approach proposed by So et al. (2024) leverages Hamilton–Jacobi reachability to address minimum-cost reach–avoid problems. Although the method is primarily formulated for deterministic systems, the authors of that work later investigate the robustness of their approach under varying levels of control and environmental noise. Their results indicate that RC-PPO achieves low cumulative cost while maintaining a comparable reach rate to other methods. Could the authors comment on how the proposed approach compares to RC-PPO in settings where stochastic noise is injected, even if only through an experimental comparison?
The work in [1] introduces reach–avoid supermartingale (RASM) certificates, which also address reach–avoid specifications in stochastic systems. These certificates combine guarantees for both safety and liveness properties and can be synthesized concurrently. It would be useful for the authors to discuss how the proposed RAPC framework compares with RASM certificates in terms of: the type of guarantees provided, the scope of applicability, and potential empirical performance differences. An experimental or conceptual comparison with this approach could further clarify the contribution of the present work.

Minor Comment On page 4, the manuscript refers to “(see in Chapter 6)”. Since this is a paper rather than a book, this should be corrected to “(see Section 6)”.

Reference: [1] Learning Control Policies for Stochastic Systems with Reach-Avoid Guarantees, Ðorde Žikelić, Mathias Lechner, Thomas A. Henzinger, and Krishnendu Chatterjee.

**Limitations:**

"yes"

**Strengths And Weaknesses:**

Strengths: The paper is generally well written and easy to follow. The authors present the core ideas in a manner that is relatively accessible, even to readers who may not be experts in the specific topic. Another strength of the work is that the proposed approach avoids the use of surrogate formulations that impose scalar trade-offs, such as reward–cost scalarization or CMDP-style cumulative cost constraints. Such surrogate formulations are commonly used in the literature on reach–avoid problems with cost minimization, but they often fail to preserve the structural properties of the original reach–avoid problem and typically require careful tuning of weights or thresholds. This tuning can be difficult and may even lead to infeasible solutions. By bypassing these surrogate approaches, the proposed framework attempts to more directly address the original reach–avoid objective.

Weaknesses: While the authors make a clear effort to provide a relevant literature review, there appear to be some omissions in the comparison with several closely related works. A more thorough discussion and comparison with existing approaches would strengthen the positioning of the paper within the broader literature. Specific examples are listed in the questions below. In addition, there are several technical aspects of the proposed method that require further clarification, which are also detailed in the questions to the authors. Finally, although the problem addressed in the paper is interesting, the manuscript would benefit from a clear motivating example or practical scenario illustrating the relevance of the problem. Including such an example could help readers better understand the practical significance and potential applications of the proposed framework.

---

> ### Author Rebuttal · Authors · 2026-03-31
>
> We thank the reviewer for the constructive feedback.
>
> # On the lack of a motivating example (Weaknesses)
> To motivate this problem, consider an Automated Guided Vehicle (AGV) navigating a warehouse. The AGV must reach a target loading dock while avoiding unpredictable collisions with human workers in a stochastic environment. The objective is to satisfy a reach-avoid probability requirement (e.g., $p \geq 0.95$) while simultaneously minimizing cumulative battery consumption. This example captures the essence of the stochastic minimum-cost reach-avoid reinforcement learning problem and we have added this example to the Introduction.
>
> # Q1: Formal guarantees of RAPCs, practical construction of RAPCs and impact of local optimality
> The Reach-Avoid Probability Certificate (RAPC) provides a formal, sufficient guarantee for the reach-avoid specification. By Definition 4.1, an RAPC $v^\pi(x)$ acts as a lower bound on the true reach-avoid probability, ensuring $\mathbb{P}\_\pi(\mathrm{RA}\_x) \geq v^\pi(x)$. Theorem 4.5 shows that for the unique fixed point $V^\pi_{g,h}$ of the proposed Bellman operator, the condition $V^\pi_{g,h}(x_0)/M \geq p$ implies that the reach-avoid requirement is satisfied with probability at least $p$. These guarantees hold under standard conditions: bounded functions $g$ and $h$, with discount factor $\gamma \in (0,1)$, which together ensure that the Bellman operator is a contraction and admits a unique fixed point (Lemma 4.4).
>
> In practice, the RAPC does not admit a closed-form representation, as $V^\pi_{g,h}$ is defined as the fixed point of a Bellman operator over continuous state spaces. To construct the certificate, the critic network learns $V^\pi_{g,h}$ via a Bellman self-consistency (temporal-difference) loss (Eq. 17--18). Note that the formal guarantee of Theorem 4.5 is stated entirely in terms of $V^\pi_{g,h}$. However, during the actual policy optimization in Algorithm 1, we maximize a surrogate objective $\frac{V^\pi_{g,h}}{\phi_\gamma^\pi}$ rather than directly enforcing the certificate condition, in order to achieve better practical performance (Fig. 6), where the compensation factor $\phi_\gamma^\pi$ is estimated from successful rollouts to stabilize the learning signal (Eq. 19). It is this surrogate formulation that introduces an approximation gap with respect to the strict reach-avoid guarantee. This trade-off is consistent with the common practice in approaches combining reinforcement learning with Control Barrier Functions, where practical performance in high-dimensional systems is prioritized over strict formal enforcement (Fig. 6).
>
> Due to the use of surrogate objectives, exact satisfaction of the original reach-avoid constraints cannot be guaranteed in general; importantly, this limitation arises from the surrogate formulation rather than local optimality. Algorithm 1 converges to a local optimum of the surrogate problem, which affects solution quality but not the nature of this relaxation. Empirically, we observe that despite the lack of global optimality guarantees, this surrogate-guided optimization consistently yields policies with strong reach-avoid performance across all evaluated benchmarks.
>
> # Q2: Comparison with RC-PPO under stochastic noise
> As shown in Fig. 5 of our submission, we evaluate RC-PPO in stochastic environments with 10% Gaussian action noise. Under these conditions, RAPCPO achieves comparable reach-avoid success rates while incurring significantly lower costs. This difference arises from the underlying modeling assumptions. RC-PPO is based on deterministic Hamilton-Jacobi reachability and does not explicitly account for stochasticity, which can lead to suboptimal cost performance when noise is present. In contrast, RAPCPO explicitly models stochastic dynamics, enabling a more principled trade-off between cost optimization and reach-avoid safety under uncertainty.
>
> # Q3: Comparison with RASM certificates
> We thank the reviewer for pointing out this highly relevant work. First, **regarding applicability and guarantees**, RASM is designed for white-box systems, relying on known analytical dynamics and structural assumptions like invariant sets [1]. While this provides strong formal guarantees, it is restrictive when dynamics are unknown. In contrast, our approach is model-free, avoids invariant-set assumptions, and learns RAPCs directly from data via temporal-difference updates. This makes it naturally compatible with standard deep RL pipelines for black-box environments.
> **Regarding empirical performance**, RASM relies on grid-based procedures, limiting it to low-dimensional systems (2-3D). Conversely, our learning-based RAPC scales to high-dimensional continuous control (e.g., 44D PointGoal). We have clarified these distinctions in the revised manuscript.
>
> *[1] Cao, Z., et al. arXiv:2512.05348 (2025).*
>
> # Minor comment:
> We will corrected "Chapter 6" to "Section 6" in the revision.

---

> > ### Author Rebuttal · Reviewer_E2mL · 2026-04-03
> >
> > I appreciate the authors for their detailed comments, clarifications, and the effort invested in presenting their work. That said, I would like to raise a concern regarding the use of the term formal guarantee. Typically, this phrase implies a rigorously established and mathematically proven property that holds over the entire system. In contrast, guarantees derived from sampling-based approximations—without explicit analysis of how the approximation error propagates to the full system—do not, in general, meet this standard of rigor. This distinction is reflected in many existing approaches that aim to provide correctness guarantees while incorporating data-driven elements. Such methods often rely on additional structural assumptions—such as Lipschitz continuity, monotonicity, or other regularity conditions—to bridge the gap between sampled behavior and global system properties. The absence of such structure in the present work makes it difficult to interpret the claimed guarantees in a fully formal sense. Moreover, the use of reinforcement learning to optimize a surrogate objective introduces an inherent trade-off: the focus shifts toward empirical performance and learning efficiency, rather than strict correctness of the specification. While Theorem 4.5 presents a sound and meaningful result—establishing a lower bound on reach-avoid satisfaction via the fixed point of a Bellman operator—its practical applicability depends on obtaining the exact solution to the Bellman equation, which is typically intractable. In practice, one must rely on sampling-based approximations, and without quantified error bounds, the formal validity of the resulting guarantee remains unclear. In this regard, incorporating structural properties of the system could be a promising direction to mitigate the trade-off between performance and guarantee correctness. Additionally, as the authors note, the use of surrogate objectives means that exact satisfaction of the original reach-avoid specification cannot, in general, be ensured. This limitation—stemming directly from the surrogate formulation—further weakens the interpretation of the results as providing formal guarantees. A more precise choice of terminology may help avoid potential confusion and better reflect the nature of the contributions.

---

> > > ### Author Response · Authors · 2026-04-06
> > >
> > > We thank the reviewer for this thoughtful and detailed follow-up comment. While we would like to respectfully clarify that the exact phrase "formal guarantee" is not used in our paper to describe the RAPCPO algorithm, we completely understand the reviewer’s broader concern. We agree that some of our summary-level phrasing may make the relationship between the theorem-level mathematical results and the practical, sampling-based algorithm appear stronger than intended.
> > >
> > > **To address this and avoid any potential confusion, we will explicitly revise the claims in our Abstract and Introduction.**
> > >
> > > Specifically:
> > > * We will remove phrases like "provably satisfying" (Line 26) and "ensuring satisfaction" (Line 69) to when referring to the RAPCPO algorithm.
> > > * Instead, we will describe RAPCPO as "optimizing a theoretically motivated surrogate objective to promote satisfaction."
> > > * In the experimental contributions (Line 95), we will change "guaranteeing satisfaction" to "empirically maintaining high satisfaction rates."
> > >
> > > These changes ensure the distinction between the exact mathematical certificate and the approximate empirical algorithm is clear throughout the text.
> > >
> > > **Regarding the scope of our theoretical and algorithmic contributions:**
> > > We would like to clarify that our work contributes at two complementary levels.
> > > 1. **At the theoretical level**, Theorem 4.5 establishes that the fixed point of the contraction-based Bellman operator yields a valid lower bound on the reach-avoid probability under exact computation. This establishes the rigorous foundation for the value/certificate construction.
> > > 2. **At the algorithmic level**, RAPCPO is a sampling-based actor-critic method built from this construction through the surrogate problems in (14) and (16). RAPCPO is theoretically motivated by Theorem 4.5, but it is not intended as an end-to-end exact guarantee for the practical learned policy under sampling and function approximation.
> > >
> > > This distinction is already reflected in the technical development: Section 4 establishes the certificate property of $V^\pi_{g,h}$, while Section 4.2 explicitly states that the normalized estimator $\hat p^\pi(x)$ is not a certified probability bound. Furthermore, Section 5 states that RAPCPO optimizes a surrogate objective and does not enforce reach-avoid feasibility at every iteration.
> > >
> > > **Regarding the absence of structural assumptions (e.g., Lipschitz continuity):**
> > > This choice is tied to our goal of developing a model-free method for black-box stochastic environments, where such quantities are typically unavailable. We agree that incorporating additional regularity assumptions to quantify approximation-error propagation would be a valuable extension. We will make this limitation and future direction more explicit in the revised manuscript.

---

### Decision · Program_Chairs · 2026-04-30

**Decision:**

Accept (regular)

**Comment:**

The problem formulation, approach and novelty is appreciated. Meanwhile, the reviewers encourage the authors to improve the literature review and better clarify contributions compared to many recent published work on reinforcement learning with reach-avoid objectives.